# Exponential Separations in Symmetric Neural Networks

**Aaron Zweig**
Courant Institute of Mathematical Sciences
New York University
az831@nyu.edu

**Joan Bruna**
Center for Data Science
New York University
bruna@cims.nyu.edu

## Abstract

In this work we demonstrate a novel separation between symmetric neural network architectures. Specifically, we consider the Relational Network [21] architecture as a natural generalization of the DeepSets [32] architecture, and study their representational gap. Under the restriction to analytic activation functions, we construct a symmetric function acting on sets of size $N$ with elements in dimension $D$, which can be efficiently approximated by the former architecture, but provably requires width exponential in $N$ and $D$ for the latter.

## 1 Introduction

The modern success of deep learning can in part be attributed to architectures that enforce appropriate invariance. Invariance to permutation of the input, i.e. treating the input as an unordered set, is a desirable property when learning *symmetric* functions in such fields as particle physics and population statistics. The simplest architectures that enforce permutation invariance treat each set element individually without allowing for interaction, as captured by the popular *DeepSet* model [18, 32].

Several architectures explicitly enable interaction between set elements, the simplest being the Relational Networks [21] that encode pairwise interaction. This may be understood as an instance of *self-attention*, the mechanism underlying Transformers [27], which have emerged as powerful generic neural network architectures to process a wide variety of data, from image patches to text to physical data. Specifically, Set Transformers [12] are special instantiations of Transformers, made permutation equivariant by omitting positional encoding of inputs, and using self-attention for pooling.

Both the DeepSets and Relational Networks architectures are universal approximators for the class of symmetric functions. But empirical evidence suggests an inherent advantage of symmetric networks using self-attention in synthetic settings [16], on point cloud data [12] and in quantum chemistry [17]. In this work, we formalize this question in terms of approximation power, and explicitly construct symmetric functions which provably require exponentially-many neurons in the DeepSets model, yet are efficiently approximated with self-attention.

This exponential separation bears notable differences from typical separation results. In particular, while the expressive power of a vanilla neural network is characterized by depth and width, expressiveness of symmetric networks is controlled particularly by *symmetric width*. In contrast to depth separations of vanilla neural networks [7], in this work we observe width separations, where the weaker architectures (even with arbitrary depth) require exponential symmetric width to match the expressive power of stronger architectures.

36th Conference on Neural Information Processing Systems (NeurIPS 2022).

**Summary of Contributions**   In this work:

- We demonstrate a *width separation* between the DeepSets and Relational Network architectures, where the former requires symmetric width $L \gg poly(N, D)$ to approximate a family of analytic symmetric functions, while the latter can approximate with polynomial efficiency. This also answers an open question of high-dimensional DeepSets representation posed in Wagstaff et al. [30]

- We introduce an extension of the Hall inner product to high dimensions that preserves low-degree orthogonality of multisymmetric powersum polynomials, which may be of independent interest.

## 2   Setup and Main Result

### 2.1   Symmetric Architectures

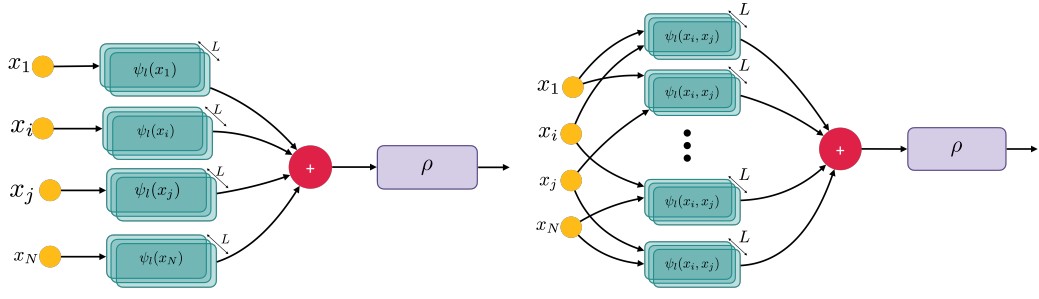

(a) DeepSets with symmetric width $L$        (b) Relational Network with symmetric width $L$

Figure 1: Architectural diagram for $\mathrm{Sym}_L$ (left) and $\mathrm{Sym}_L^2$ (right)

To introduce the symmetric architectures, we must first characterize how to treat sets as inputs. We will consider sets of size $N$, where each element of the set is a vector of dimension $D$. In particular, we will represent a set as a matrix $X \in \mathbb{C}^{D \times N}$. Thus, each column vector $x_n \in \mathbb{C}^D$ is an element of the set. Note that we consider complex-valued inputs because the natural inner product over symmetric polynomials integrates over the complex unit circle, see Macdonald [14] or Theorem 4.3.

A function $f : \mathbb{C}^{D \times N} \to \mathbb{C}$ is *symmetric* if $f(X) = f(X\Pi)$ for any permutation matrix $\Pi \in \mathbb{R}^{N \times N}$, i.e. if $f$ is invariant to permuting the columns of $X$. In other words, a symmetric function treats the input $X$ as an unordered set of column vectors. Given the *symmetric width* parameter $L$, we consider two primary symmetric architectures:

**Definition 2.1.** *Let $Sym_L$ denote the class of* singleton symmetric networks *with symmetric width L, i.e. functions $f$ of the form:*

$$f(X) = \rho(\phi_1(X), \dots, \phi_L(X)) \tag{1}$$

$$\phi_l(X) = \sum_{n=1}^{N} \psi_l(x_n) \tag{2}$$

*where $\{\psi_l : \mathbb{C}^D \to \mathbb{C}\}_{l=1}^L$ and $\rho : \mathbb{C}^L \to \mathbb{C}$ are arbitrary neural networks with analytic activations.*

The class $\mathrm{Sym}_L$ is exactly the architecture of DeepSets [32] restricted to analytic activations. However, we introduce this notation to differentiate this class from the more expressive architectures that allow for pairwise interaction among set elements.

From the theory of symmetric polynomials, if $L \geq L^* := \binom{N+D}{N} - 1$, then $f \in \mathrm{Sym}_L$ is a universal approximator for any analytic symmetric function [19]. Therefore we will primarily be interested in the expressive power of $\mathrm{Sym}_L$ for $L < L^*$.

**Definition 2.2.** *Let $Sym_L^2$ denote the class of* pairwise symmetric networks *with symmetric width L, i.e. functions $f$ of the form:*

$$f(X) = \rho(\phi_1(X), \ldots, \phi_L(X)) \tag{3}$$

$$\phi_l(X) = \sum_{n,n'=1}^{N} \psi_l(x_n, x_{n'}) \tag{4}$$

*where $\{\psi_l : \mathbb{C}^{D \times D} \to \mathbb{C}\}_{l=1}^{L}$ and $\rho : \mathbb{C}^L \to \mathbb{C}$ are arbitrary neural networks with analytic activations.*

Similarly, the class $Sym_L^2$ is exactly the architecture of Relational Pooling [21] with analytic activations. We note this architecture is also equivalent to the 2-ary instantiation of Janossy Pooling [16].

## 2.2 Main Result

Our main result demonstrates an exponential separation, where $Sym_L$ requires exponentially large symmetric width $L$ to match the expressive power of the class $Sym_L^2$ for $L = 1$. We choose norms to make this separation as prominent as possible: there is a hard function that can be approximated in $Sym_L^2$ in the infinity norm, but cannot be approximated in $Sym_L$ even in an appropriately chosen $L_2$ norm with respect to some non-trivial data distribution.

We require one activation assumption to realize the $Sym_L^2$ approximation:

**Assumption 2.3.** *The activation $\sigma : \mathbb{C} \to \mathbb{C}$ is analytic, and for a fixed $D, N$ there exist two-layer neural networks $f_1, f_2$ using $\sigma$, both with $O\left(D^2 + D \log \frac{D}{\epsilon}\right)$ width and $O(D \log D)$ bounded weights, such that:*

$$\sup_{|\xi| \le 3} |f_1(\xi) - \xi^2| \le \epsilon, \qquad \sup_{|\xi| \le 3} \left| f_2(\xi) - \left(1 - (\xi/4)^{\min(D, \sqrt{N}/2)}\right) \frac{\xi - 1/4}{\xi/4 - 1} \right| \le \epsilon \tag{5}$$

Essentially this assumption guarantees that networks built with the analytic activation $\sigma$ are able to efficiently approximate the map $\xi \to \xi^2$, and, a truncated form of the finite Blaschke product[8] with one zero at $\xi = 4$. We show in Lemma G.3 that the $\exp$ activation satisfies this assumption.

**Theorem 2.4** (Exponential width-separation). *Fix $N$ and $D > 1$, and a non-trivial data distribution $\mu$ on $D \times N$ copies of the unit complex circle $(S^1)^{D \times N}$.*

*Then there exists an analytic symmetric function $g : \mathbb{C}^{D \times N} \to \mathbb{C}$ such that $\|g\|_{L_2(\mu)} = 1$ and:*

- *For $L \le N^{-2} \exp(O(\min(D, \sqrt{N})))$,*

$$\min_{f \in Sym_L} \|f - g\|_{L_2(\mu)}^2 \ge \frac{1}{12} . \tag{6}$$

- *There exists $f \in Sym_L^2$ with $L = 1$, parameterized with an activation $\sigma$ that satisfies Assumption 2.3, with width $poly(N, D, 1/\epsilon)$, depth $O(\log D)$, and max weight $O(D \log D)$ such that over $(S^1)^{D \times N}$:*

$$\|f - g\|_\infty \le \epsilon \tag{7}$$

**Remark 1.** *The lower bound is completely independent of the width and depth of the parameterized networks $\{\psi_l\}$ and $\rho$. The only parameter that the theorem restricts is the symmetric width $L$. This is in sharp contrast to the separations of vanilla networks [7], where there is a natural trade-off between width and depth.*

**Remark 2.** *In the upper bound, we consider the network $f \in Sym_L^2$ to have width and depth in the usual sense of vanilla neural networks, where the parameterized maps $\{\psi_l\}$ and $\rho$ obey the width, depth, and weight bounds given.*

# 3   Related Work

## 3.1   Depth Separation

Numerous works have studied the difference in expressive power between different neural network architectures. Many of these works center on the representational gap between two-layer and three-layer networks [4, 7]. In particular, recent works have focused on generalizing the family of functions that realize these separations, to various radial functions [20] and non-radial functions [28].

A separate line of work considers separations between networks when the depth varies polynomially [24]. Notably, Vardi, Yehudai, and Shamir [26] demonstrates that depth has a greater impact on expressivity than width, in the case of vanilla neural networks.

## 3.2   Symmetric Architectures

We primarily consider the symmetric neural network parameterization as introduced in DeepSets[32], with PointNet[18] a similar symmetric parameterization using a different pooling function. Simple linear equivariant layers were also introduced in Zaheer et al. [32].

In the context of relationships between objects in an image, the first symmetric architecture enabling explicit pairwise interaction was introduced in Santoro et al. [21]. More complicated symmetric architectures, allowing for higher-order interaction and more substantial equivariant layers, were built on top of attention primitives [12, 13]. And the notion of explicit high-order interactions between set elements before symmetrizing is formalized in the architecture of Janossy pooling [16].

Symmetric architectures are generalized by graph neural networks [10, 22], under the restriction to the complete graph.

## 3.3   Symmetric Network Expressivity

The dependence of representational power on the symmetric width parameter $L$ was first demonstrated in the $D = 1$ case. Under the strong condition $L < N$, it was proven there are symmetric functions which cannot be exactly represented by a DeepSets network [29], and this was later strengthened to functions which cannot be approximated in the infinity norm to arbitrary precision [30].

The work introducing Janossy pooling [16] also includes a theoretical result showing singleton networks cannot exactly represent some particular pairwise symmetric network. Crucially however, this result is restricted to a simplified, non-universal symmetric architecture excluding the $\rho$ transformation, and therefore does not characterize the real-world architectures given above.

The question of expressiveness in symmetric networks may also be generalized to graph neural networks, with a focus on distinguishing non-isomorphic graphs as compared to the Weissfeler-Lehman test[31] and calculating invariants such as substructure counting[3]. In particular, one may understand expressiveness in symmetric networks incorporating pairwise interaction as the ability to learn functions of the complete graph decorated with edge features.

## 3.4   Symmetric Polynomial Theory

Our proofs rely on the technical machinery of symmetric polynomial theory, thoroughly characterized in Macdonald [14]. In particular, we utilize the integral representation of the finite-variable Hall Inner product as introduced in Section A. Because this integral is defined over the complex unit circle, we consequently consider complex-valued neural networks [1].

The connection of symmetric networks to the powersum polynomials was first observed in Zaheer et al. [32], and likewise the multisymmetric powersum polynomials have been applied in higher dimensional

symmetric problems [15, 23]. The algebraic properties of the multisymmetric powersum polynomials are well-studied, for example as a basis of higher dimensional symmetric polynomials [19] and through their algebraic dependencies [6]. However, to the best of our knowledge this is the first work to apply the Hall inner product to symmetric neural networks, and to extend this inner product to yield low-degree orthogonality over the multisymmetric polynomials.

# 4 Warmup: One-dimensional set elements

To begin, we consider the simpler case where $D = 1$, i.e. where we learn a symmetric function acting on a set of scalars. It was already observed in Zaheer et al. [32] that the universality of DeepSets could be demonstrated by approximating the network with symmetric polynomials. We first demonstrate that through this approximation, we can relate the symmetric width $L$ to expressive power.

## 4.1 Symmetric Polynomials

In order to approximate symmetric networks by symmetric polynomials, we choose a suitable basis. The powersum polynomials serve as the natural choice, as their structure matches that of a singleton symmetric network, and they obey very nice orthogonality properties that we detail below.

**Definition 4.1.** *For $k \in \mathbb{N}$ and $x \in \mathbb{C}^N$, the* normalized powersum polynomial *is defined as*

$$p_k(x) = \frac{1}{\sqrt{k}} \sum_{n=1}^{N} x_n^k$$

*with $p_0(x) = 1$.*

A classical result in symmetric polynomial theory is the existence of an $L_2$ inner product that grants orthogonality for products of powersums. To make this notion explicit and keep track of products, we index products with partitions.

**Definition 4.2.** *An* integer partition $\lambda$ *is non-increasing, finite sequence of positive integers $\lambda_1 \geq \lambda_2 \geq \cdots \geq \lambda_k$. The weight of the partition is given by $|\lambda| = \sum_{i=1}^{k} \lambda_i$. The length of a partition $l(\lambda)$ is the number of terms in the sequence.*

Then we characterize a product of powersums by:

$$p_\lambda(x) = \prod_i p_{\lambda_i}(x) \tag{8}$$

This notation intentionally also allows for the empty partition, such that if $\lambda = \varnothing$ then $p_\lambda = 1$. All together, we can now state the following remarkable fact:

**Theorem 4.3** ([14, Chapter VI (9.10)] )**.** *There exists a $L_2(d\nu)$ inner product (for some probability measure $\nu$) such that, for partitions $\lambda, \mu$ with $|\lambda| \leq N$:*

$$\langle p_\lambda, p_\mu \rangle_V = z_\lambda \mathbb{1}_{\lambda = \mu} \tag{9}$$

*where $z_\lambda$ is some combinatorial constant.*

We index this inner product with $V$ because it is written as an expectation with respect to a density proportional to the squared Vandermonde polynomial (see Section A for the precise definition). This inner product may also be considered the finite-variable specialization of the Hall inner product, defined on symmetric polynomials over infinitely many variables [14, Chapter I (4.5)].

It's easy to check that the degree of $p_\lambda$ is equal to $|\lambda|$. So this theorem states that the powersum terms $p_\lambda$ are "almost" an orthogonal basis, except for correlation between two high-degree terms.

Let us remark that we assume analytic activations for the sake of this theorem, as the orthogonality property does not hold for symmetric polynomials with negative exponents. However, in exchange for that assumption we can apply this very powerful inner product, that ultimately results in the irrelevance of network depth.

## 4.2 Projection Lemma

Before we can proceed to prove a representational lower bound, we need one tool to better understand $f \in \text{Sym}_L$. Utilizing the orthogonality properties of the inner product $\langle \cdot, \cdot \rangle_V$ allows us to project any $f \in \text{Sym}_L$ to a simplified form, while keeping a straightforward dependence on $L$.

For example, consider some uniformly convergent power series (with no constant term) $\phi(x) = \sum_{i=1}^{\infty} c_{ik} p_k(x)$. We claim $\langle p_2 p_1, \phi^3 \rangle_V = 0$. Indeed, expanding $\phi^3$, one exclusively gets terms of the form $p_{k_1} p_{k_2} p_{k_3}$, and because the partition $\{k_1, k_2, k_3\}$ is of a different length than $\{2, 1\}$, they are clearly distinct partitions so by orthogonality $\langle p_2 p_1, p_{k_1} p_{k_2} p_{k_3} \rangle_V = 0$.

Motivated by this observation, we can project $f$ to only contain products of two terms. Let us introduce $\mathcal{P}_1$ to be the orthogonal projection onto $span(\{p_t : 1 \leq t \leq N/2\})$, and $\mathcal{P}_2$ to be the orthogonal projection onto $span(\{p_t p_{t'} : 1 \leq t, t' \leq N/2\})$.

**Lemma 4.4.** *Given any $f \in \text{Sym}_L$, we may choose coefficients $v_{ij}$ over $i \leq j \leq L$, and symmetric polynomials $\phi_i$ over $i \leq L$, such that:*

$$\mathcal{P}_2 f = \sum_{i \leq j}^{L} v_{ij} (\mathcal{P}_1 \phi_i)(\mathcal{P}_2 \phi_j) \tag{10}$$

## 4.3 Rank Lemma

Given the reduced form of $f$ above, we may now go about lower bounding its approximation error to a given function $g$.

By the properties of orthogonal projection, we have $\|f - g\|_V^2 \geq \|\mathcal{P}_2(f - g)\|_V^2$. And by Parseval's theorem, the function approximation error $\|\mathcal{P}_2 f - \mathcal{P}_2 g\|_V^2$ equals

$$\sum_{t \leq t'} \left( \left\langle \mathcal{P}_2 f, \frac{p_t p_{t'}}{\|p_t p_{t'}\|_V} \right\rangle_V - \left\langle \mathcal{P}_2 g, \frac{p_t p_{t'}}{\|p_t p_{t'}\|_V} \right\rangle_V \right)^2 .$$

Rearranging the orthogonal coefficients in the form of matrices, we have the following fact:

**Lemma 4.5.** *Given any $f \in \text{Sym}_L$, and $g$ such that $P_2 g = g$, we have the bound*

$$\|\mathcal{P}_2 f - \mathcal{P}_2 g\|_V^2 \geq \frac{1}{2} \|F - G\|_F^2 \tag{11}$$

*where $F, G \in \mathbb{C}^{N/2 \times N/2}$ are matrices with entries $F_{tt'} = \langle \mathcal{P}_2 f, p_t p_{t'} \rangle_V$, $G_{tt'} = \langle \mathcal{P}_2 g, p_t p_{t'} \rangle_V$. Furthermore, $F$ has maximum rank $L$.*

The significance of this lemma is the rank constraint: it implies that choosing symmetric width $L$ corresponds to a maximum rank $L$ on the matrix $F$. From here, we can use standard arguments about low-rank approximation in the Frobenius norm to yield a lower bound.

## 4.4 Separation in one-dimensional case

Our main goal in this section is to construct a hard symmetric function $g$ that cannot be efficiently approximated by $\text{Sym}_L$ for $L \leq N/4$. It is not particularly expensive for the symmetric width $L$ to scale linearly with the set size $N$: however, we will use the same proof structure to prove Theorem 2.4, which will require $L$ to scale exponentially.

**Theorem 4.6.** *For $D = 1$:*

$$\max_{\|g\|_V = 1} \min_{f \in \text{Sym}_L} \|f - g\|_V^2 \geq 1 - \frac{2L}{N} \tag{12}$$

*In particular, for $L = \frac{N}{4}$ we recover a constant lower bound of $\frac{1}{2}$.*

*Proof (sketch).* Choose $g$ such that $\mathcal{P}_2 g = g$. Then because $\mathcal{P}_2$ is an orthogonal projection and applying Lemma 4.5:

$$\min_{f \in \mathrm{Sym}_L} \|f - g\|_V^2 \geq \min_{f \in \mathrm{Sym}_L} \|\mathcal{P}_2 f - \mathcal{P}_2 g\|_V^2 \tag{13}$$

$$\geq \frac{1}{2} \min_{\mathrm{rank}(F) \leq L} \|F - G\|_F^2 \tag{14}$$

We note that $\|p_t p_t\|_V^2 = z_{\{t,t\}} = 2$, so the choice of $g = \frac{1}{\sqrt{N}} \sum_{t=1}^{N/2} p_t p_t$ can be seen to obey $\|g\|_V = 1$, and implies that $G$ is the scaled identity matrix $\frac{2}{\sqrt{N}} I \in \mathbb{C}^{N/2 \times N/2}$. Then by standard properties of the SVD:

$$\min_{f \in \mathrm{Sym}_L} \|f - g\|_V^2 \geq \frac{1}{2} \min_{\mathrm{rank}(F) \leq L} \left\|F - \frac{2}{\sqrt{N}} I\right\|_F^2 \tag{15}$$

$$= \frac{1}{N/2} \min_{\mathrm{rank}(F) \leq L} \|F - I\|_F^2 \tag{16}$$

$$= \frac{1}{N/2} (N/2 - L) \tag{17}$$

$$= 1 - \frac{2L}{N} \tag{18}$$

$\square$

# 5 Proof Sketch of Main Result

## 5.1 Challenges for High-dimensional Set Elements

We'd like to strengthen this separation in several ways:

- Generalize to the $D > 1$ case,

- Realize a separation where the symmetric width $L$ must scale exponentially in $N$ and $D$, showing that $\mathrm{Sym}_L$ is infeasible,

- Show the hard function $g$ can nevertheless be efficiently approximated in $\mathrm{Sym}_L^2$ for $L$ polynomial in $N$ and $D$

First, in order to approximate via polynomials in the high-dimenionsal case, we will require the high-dimensional analogue of powersum polynomials:

**Definition 5.1.** *For a multi-index $\alpha \in \mathbb{N}^D$, the* normalized multisymmetric powersum polynomial *is defined as:*

$$\mathbf{p}_\alpha(X) = \frac{1}{\sqrt{|\alpha|}} \sum_n \prod_d x_{dn}^{\alpha_d} . \tag{19}$$

So the plan is to find a high-dimensional analogue of Lemma 4.4 and Lemma 4.5, now using multisymmetric powersum polynomials, mimic the proof of the $D = 1$ case, and then additionally show the hard function $g$ is efficiently computable in the pairwise symmetric architecture. Note that because the algebraic basis of multisymmetric powersum polynomials is of size $L^* = \binom{N+D}{N} - 1$, we can expect an exponential separation when we apply a similar rank argument.[1]

---

[1] We subtract one in order to discount the constant polynomial.

## 5.2 Sketch of Main result (lower bound)

Because we are in high dimensions, we cannot simply apply the restricted Hall inner product introduced in Theorem 4.3. To the best of our knowledge, there is no standard generalization of the Hall inner product to multi-symmetric polynomials that preserves the orthogonality property. For the main technical ingredient in the high-dimensional case we introduce a novel generalization, which builds on two inner products.

First, we introduce a new input distribution $\nu$ over set inputs $X \in \mathbb{C}^{D \times N}$, and induce an $L_2$ inner product:

$$\langle f, g \rangle_{\mathcal{A}} = \mathbb{E}_{X \sim \nu} \left[ f(X) \overline{g(X)} \right] . \tag{20}$$

We use this inner product to measure the approximation error of $\text{Sym}_L$. That is, we seek a lower bound to $\min_{f \in \text{Sym}_L} \| f - g \|_{\mathcal{A}}$, for a suitable choice of hard function $g$.

We can now apply an analogue of Lemma 4.4 to project $f$ to a simplified form. But we cannot immediately apply an analogue of Lemma 4.5, as it relied on Parseval's theorem and the low-degree multisymmetric powersum polynomials are not orthogonal in this inner product. Put another way, if we represent $\langle \cdot, \cdot \rangle_{\mathcal{A}}$ as a matrix in the basis of low-degree multisymmetric powersums, it will be positive-definite but include some off-diagonal terms.

The idea is to now introduce a new inner product with a different input distribution $\nu_0$

$$\langle f, g \rangle_{\mathcal{A}_0} = \mathbb{E}_{X \sim \nu_0} \left[ f(X) \overline{g(X)} \right] , \tag{21}$$

and define the bilinear form

$$\langle f, g \rangle_* = \langle f, g \rangle_{\mathcal{A}} - 2 \langle f, g \rangle_{\mathcal{A}_0} . \tag{22}$$

Typically positive-definiteness is lost when subtracting two inner products, but we prove that $\langle \cdot, \cdot \rangle_*$ is an inner product when restricted to a particular subspace of symmetric polynomials (see Theorem D.3). Furthermore, the careful choice of $\nu$ and $\nu_0$ cancels the off-diagonal correlation of different multisymmetric powersums, so they are orthogonal under this new inner product $\langle \cdot, \cdot \rangle_*$.

By the norm domination $\| \cdot \|_{\mathcal{A}} \geq \| \cdot \|_*$, we are able to pass from the former $L_2$ norm to the latter norm that obeys orthogonality, and apply an analogue of the Rank Lemma 4.5. Thus we derive a lower bound using any hard function $g$ whose corresponding matrix $G$ (built from orthogonal coefficients) is diagonal and high-rank. And because the total number of polynomials is $L^*$, the rank argument now yields an exponential separation.

Based on this proof, we have much freedom in our choice of $g$. By choosing its coefficients in the basis of multisymmetric powersum polynomials, it's easy to enforce the conditions that $G$ is diagonal and high-rank for variety of possible functions. However, ensuring that $g$ is not pathological (i.e. that it is bounded and Lipschitz), and can be efficiently approximated in $\text{Sym}_L^2$, requires a more careful choice.

## 5.3 Sketch of Main Result (upper bound)

It remains to approximate the hard function $g$ with a network from $\text{Sym}_L^2$. First we must make a choice of $g$ in particular.

Based on the lower bound proof, the desiderata for $g$ is that it is supported exclusively on terms of the form $\mathbf{p}_\alpha \mathbf{p}_\alpha$ over many values of $\alpha$, as this induces a diagonal and high-rank matrix $G$ in an analogue of Lemma 4.5. Furthermore, by simple algebra one can confirm that $\mathbf{p}_\alpha(X)\mathbf{p}_\alpha(X) = \frac{1}{|\alpha|} \sum_{n,n'} \prod_{d=1}^{D} (x_{dn} x_{dn'})^{\alpha_d}$, so $g$ supported on these polynomials can clearly be written in the form of a network in $\text{Sym}_L^2$. This structure of $g$ guarantees difficult approximation, and is akin to the radial structure of the hard functions introduced in works on depth separation [7].

We must however be careful in our choice of $g$: for the matrix $G$ to be high-rank, $g$ must be supported on exponentially many powersum polynomials. But this could make $\|g\|_\infty$ exponentially large, and therefore challenging to approximate efficiently with a network from $\mathrm{Sym}_L^2$.

We handle this difficulty by defining $g$ in a different way. We introduce a finite Blaschke product $\mu(\xi) = \frac{\xi - 1/4}{\xi/4 - 1}$, a function that analytically maps the unit complex circle to itself. Then the choice

$$g(X) = \sum_{n,n'=1}^{N} \prod_{d=1}^{D} \mu(x_{dn} x_{dn'}) \tag{23}$$

ensures that $\|g\|_\infty$, $\|g\|_{\mathcal{A}}$, and $\mathrm{Lip}(g)$ are all polynomial in $N, D, \frac{1}{\epsilon}$ for $\epsilon$ approximation error (see Lemma E.3). Furthermore, again from simple algebra it is clear that $g$ is only supported on terms of the form $\mathbf{p}_\alpha \mathbf{p}_\alpha$. So it remains to show that the induced diagonal matrix $G$ is effectively high rank, which follows from expanding the Blaschke products.

Satisfied that this choice of $g$ will meet the desiderata for the lower bound, and has no pathological behavior, it remains to construct $f \in \mathrm{Sym}_L^2$ for $L = 1$ that approximates $g$. That is, choose $\psi_1$ and $\rho$ so that $g(X) \approx \rho\left(\sum_{n,n'=1}^{N} \psi_1(x_n, x_{n'})\right)$. Clearly we may take $\rho$ to be the identity, and $\psi_1(x_n, x_{n'})$ to approximate $\prod_{d=1}^{D} \mu(x_{dn} x_{dn'})$, which is straightforwardly calculated in depth $O(\log D)$ by performing successive multiplications in a binary-tree like structure (see Theorem F.1).

Ultimately, we use a slight variant of this function for the formal proof. Because the orthogonality of our newly introduced inner product $\langle \cdot, \cdot \rangle_*$ only holds for low-degree polynomials, we must truncate high-degree terms of $g$; we confirm in Appendix F that this truncation nevertheless preserves the properties we care about.

## 6   Discussion

In this work, we've demonstrated how symmetric width captures more of the expressive power of symmetric networks than depth when restricted to analytic activations, by evincing an exponential separation between two of the most common architectures that enforce permutation invariance.

The most unusual property of this result is the complete independence of depth, owing to the unique orthogonality properties of the restricted Hall inner product when paired with the assumption of analyticity. This stands in contrast to the case of vanilla neural networks, for which separations beyond small depth would resolve open questions in circuit complexity suspected to be quite hard [25]. Furthermore, the greater dependence on width than depth is a unique property to symmetric networks, whereas the opposite is true for vanilla networks [26].

A natural extension would be to consider the simple equivariant layers introduced in Zaheer et al. [32], which we suspect will not substantially improve approximation power of $\mathrm{Sym}_L$. Furthermore, allowing for multiple such equivariant layers, this network becomes exactly akin to a Graph Convolutional Network [10] on a complete graph, whereas $\mathrm{Sym}_L^2$ corresponds to a message passing network [9] as it is capable of interpreting edge features.

### 6.1   Limitations

The major limitation of this result is the restriction to analytic functions. Although analytic symmetric functions nevertheless appear crucially in the study of exactly solvable quantum systems [2, 11], this assumption may be be overly strict for general problems of learning symmetric functions. We nevertheless conjecture that these bounds will still hold even allowing for non-analytic activations, and consider this an exciting question for future work. Additionally, whether the hard function $g$ can be efficiently learned with gradient descent remains unclear, and future work could touch on the learnability.

**Acknowledgements:** This work has been partially supported by the Alfred P. Sloan Foundation, NSF RI-1816753, NSF CAREER CIF-1845360, and NSF CCF-1814524.

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
