# Appendices

# A    Preliminaries

## A.1    Notation

We'll use $\mathbb{N}$ to denote the naturals including $0$. The indicator function for the condition $x = y$ is written as $\mathbb{1}_{x=y}$. Given an integer *weak composition* $\alpha \in \mathbb{N}^D$, we will often consider the multidimensional polynomial $z^\alpha = \prod_{d=1}^{D} z_d^{\alpha_d}$. For two vectors $x, x' \in \mathbb{C}^D$, we denote their elementwise product by $x \circ x'$.

## A.2    Inner Products

We introduce two $L_2$ inner products (defined with respect to probability measures) we'll use throughout the work. For symmetric functions $f, g : \mathbb{C}^N \to \mathbb{C}$, define:

$$\langle f, g \rangle_V = \frac{1}{(2\pi)^N N!} \int_{[0,2\pi]^N} f(e^{i\boldsymbol{\theta}})\overline{g(e^{i\boldsymbol{\theta}})}|V(e^{i\boldsymbol{\theta}})|^2 d\boldsymbol{\theta} \,, \tag{24}$$

where for $z \in \mathbb{C}^N$, we have the Vandermonde determinant

$$V(z) = \prod_{1 \leq i < j \leq N} (z_j - z_i) \,. \tag{25}$$

This inner product is well-known in the theory of symmetric polynomials, as a finite-variable analogue of the Hall inner product [14]. Equivalently, if we let $V$ denote the joint density of eigenvalues of a Haar-distributed unitary matrix in $\mathbb{C}^{N \times N}$, it is known [5] that this inner product may be written as

$$\langle f, g \rangle_V = \mathbb{E}_{y \sim V} \left[ f(y)\overline{g(y)} \right] \,. \tag{26}$$

For arbitrary functions $f, g : \mathbb{C}^D \to \mathbb{C}$, we also consider the $L_2$ inner product given as an expectation over $D$ random variables

$$\langle f, g \rangle_{S^1} = \frac{1}{(2\pi)^D} \int_{[0,2\pi]^D} f(e^{i\boldsymbol{\theta}})\overline{g(e^{i\boldsymbol{\theta}})}d\boldsymbol{\theta} \tag{27}$$

$$= \mathbb{E}_{q \sim (S^1)^D} \left[ f(q)\overline{g(q)} \right] \,, \tag{28}$$

with the notation $q \sim (S^1)^D$ meaning each entry of $q$ is i.i.d. uniform on $S^1$.

For this inner product, we will introduce the following notation. For a multi-index $\alpha \in \mathbb{N}^D$ and a dummy variable $q$ of dimension $D$, we let $q^\alpha$ denote the polynomial function $z \mapsto z^\alpha$. Then it's clear that

$$\langle q^\alpha, q^\beta \rangle_{S^1} = \mathbb{1}_{\alpha=\beta} \,. \tag{29}$$

Note that we will consider this inner product over varying dimensions throughout the paper, but it will be clear from context the dimension, i.e. how many i.i.d. random variables uniform on $S^1$ we are sampling over.

## A.3    Symmetric Polynomials

We remind the notation from the main body: $p_0(x) = 1$, and for $k \in \mathbb{N} \setminus \{0\}$ and any partition $\lambda$:

$$p_k(x) = \frac{1}{\sqrt{k}} \sum_{n=1}^{N} x_n^k \tag{30}$$

$$p_\lambda(x) = \prod_i p_{\lambda_i}(x) \,. \tag{31}$$

We will also sometimes use set notation to index products of powersums. For example, $p_{\{2,1\}} = p_2 p_1 = p_1 p_2$.

Finally, we need the notation that if $n_t$ denotes the number of times $t$ appears in $\lambda$, then $z_\lambda = \prod_t n_t!$. Note that this definition of $z_\lambda$ is slightly different that most texts, as we're considering the normalized powersums.

Then we can state Theorem 4.3 explicitly:

**Theorem A.1.** *[[14, Chapter VI (9.10)] ] For partitions $\lambda, \mu$ with $|\lambda| \leq N$:*

$$\langle p_\lambda, p_\mu \rangle_V = z_\lambda \mathbb{1}_{\lambda=\mu} \,. \tag{32}$$

### A.4  Multisymmetric Polynomials

When $D > 1$, in order to approximate our network with polynomials, we introduce the multivariate analog of symmetric polynomials. For example, suppose $D = 2$, and we write our set elements the following way:

$$X = \left\{ \begin{bmatrix} y_1 \\ z_1 \end{bmatrix}, \begin{bmatrix} y_2 \\ z_2 \end{bmatrix}, \dots \begin{bmatrix} y_N \\ z_N \end{bmatrix} \right\}$$

Then a basis of symmetric functions is given by the multisymmetric power sum polynomials, some examples:

$$\mathbf{P}_{(2,3)}(X) = \frac{1}{\sqrt{2+3}} \sum_n y_n^2 z_n^3 \tag{33}$$

$$\mathbf{P}_{(4,1)}(X) = \frac{1}{\sqrt{4+1}} \sum_n y_n^4 z_n^1 \,. \tag{34}$$

For general $N$ and $D$, our input is $X \in \mathbb{C}^{D \times N}$ where we want functions that are invariant to permuting the columns $x_n$ of this matrix. Note that we write scalar entries of this matrix as $x_{dn}$.

**Definition A.2.** *For a multi-index $\alpha \in \mathbb{N}^D$, the* normalized multisymmetric powersum polynomial *is defined as:*

$$\mathbf{p}_\alpha(X) = \frac{1}{\sqrt{|\alpha|}} \sum_n x_n^\alpha \tag{35}$$

$$= \frac{1}{\sqrt{|\alpha|}} \sum_n \prod_d x_{dn}^{\alpha_d} \tag{36}$$

*with $\mathbf{p}_0 = 1$.*

An algebraic basis of symmetric functions in this setting is given by all $\mathbf{p}_\alpha$ for all $|\alpha| \leq N$, where $|\alpha| = \sum_d \alpha_d$ (for a proof see Rydh [19]).

We remind the notation from the introduction, where $L^*(N, D) = |\{\alpha \in \mathbb{N}^D : |\alpha| \leq N\}| = \binom{N+D}{N} - 1$ is the size of this algebraic basis (discouting the constant polynomial). Intuitively then it's clear why $L \geq L^*$ will make $\text{Sym}_L$ a universal approximator, as each of the $L$ symmetric features $\{\phi_l\}_{l=1}^L$ will calculate one of these basis elements.

## B  One Dimensional Set Elements

We will first consider the setting where $D = 1$, i.e. each set element is a scalar. In this setting, we will amend notation slightly so that we consider symmetric functions $f$ acting on $x \in \mathbb{C}^N$, where each $x_n$ is a scalar set element.

## B.1 Projection Lemma

Let us remind $\mathcal{P}_1$ to be the orthogonal projection onto $span(\{p_t : 1 \le t \le N/2\})$, and $\mathcal{P}_2$ to be the orthogonal projection onto $span(\{p_t p_{t'} : 1 \le t, t' \le N/2\})$.

**Lemma B.1.** *Given any $f \in Sym_L$, we may choose coefficients $v_{ij}$ over $i \le j \le L$, and symmetric polynomials $\phi_i$ over $i \le L$, such that:*

$$\mathcal{P}_2 f = \sum_{i \le j}^{L} v_{ij} (\mathcal{P}_1 \phi_i)(\mathcal{P}_2 \phi_j) . \tag{37}$$

*Proof.* Consider the general parameterization of $f$ given in Equation 1. Because all network activations are analytic, we can write all maps parameterizing $f$ by power series.

Note that the inner product $\langle \cdot, \cdot \rangle_V$ integrates over a compact domain, therefore the projection $\mathcal{P}_2 f$ will be determined by the value of $f$ restricted to that domain. Thus, all power series in the sequel will converge uniformly and we may freely interchange infinite sums with each other as well as with inner products.

Explicitly, to parameterize $f$ we write $\psi_l(x_n) = c_{l0} + \sum_{k=1}^{\infty} \frac{c_{lk}}{\sqrt{k}} x_n^k$ so that $\phi_l(x) = \sum_{n=1}^{N} \psi_l(x_n) = N c_{l0} + \sum_{k=1}^{\infty} c_{lk} p_k(x)$.

Because $\rho$ is also given as a power series, it can be equivalently written as a power series with all variables having constant offsets. So we can subtract the constant terms from every $\phi_l$ and write:

$$\rho(y) = \sum_{\eta \in \mathbb{N}^L} v_\eta y^\eta , \tag{38}$$

$$\phi_l = \sum_{k=1}^{\infty} c_{lk} p_k , \tag{39}$$

where $y^\eta = \prod_{n=1}^{N} y_n^{\eta_n}$. Hence

$$f = \rho(\phi_1, \dots, \phi_L) = \sum_\eta v_\eta \phi^\eta . \tag{40}$$

We proceed to calculate $\mathcal{P}_2 f$. To begin, consider $\langle p_t p_{t'}, \phi^\eta \rangle$ for any choice of indices $1 \le t, t' \le N/2$. To illustrate, suppose $\eta_i = \eta_j = \eta_k = 1$ and $\eta$ is 0 everywhere else. Then we may write

$$\langle p_t p_{t'}, \phi^\eta \rangle_V = \langle p_t p_{t'}, \phi_i \phi_j \phi_k \rangle_V = \sum_{i'=1}^{\infty} \sum_{j'=1}^{\infty} \sum_{k'=1}^{\infty} c_{ii'} c_{jj'} c_{kk'} \langle p_t p_{t'}, p_{i'} p_{j'} p_{k'} \rangle_V = 0 . \tag{41}$$

In other words, after distributing the product $\phi_i \phi_j \phi_k$, we are left with a sum of terms of the form $p_{i'} p_{j'} p_{k'}$. So treated as partitions, we clearly have $\{i', j', k'\} \ne \{t, t'\}$, where all these indices are positive. Thus, because $t + t' \le N$, we can apply the orthogonality property of the inner product to conclude $\langle p_t p_{t'}, p_{i'} p_{j'} p_{k'} \rangle_V = 0$.

By similar logic, $\langle p_t p_{t'}, \phi^\eta \rangle = 0$ whenever $|\eta| \ne 2$, so we may cancel all such terms in the expansion of $f$ to get

$$\mathcal{P}_2 f = \mathcal{P}_2 \left( \sum_{\eta \in \mathbb{N}^L} v_\eta \phi^\eta \right) = \sum_{|\eta|=2} v_\eta \mathcal{P}_2 \phi^\eta .$$

Here we can simplify notation. Let $\{e_i\}_{i=1}^{L}$ denote the standard basis vectors in dimension $L$. Every $\eta \in \mathbb{N}^L$ with $|\eta| = 2$ can be written as $\eta = e_i + e_j$, so let $v_{ij} := v_{e_i + e_j}$. Then we can rewrite:

$$\mathcal{P}_2 f = \sum_{i \le j}^{L} v_{ij} \mathcal{P}_2 \phi_i \phi_j .$$

Finally, note again by orthogonality we have that $\mathcal{P}_2(p_{i'}p_{j'}) = 0$ if it is not the case that $1 \leq i', j' \leq N/2$. So observe that we may pass from $\mathcal{P}_2$ to $\mathcal{P}_1$:

$$\mathcal{P}_2\phi_i\phi_j = \mathcal{P}_2\left(\sum_{i'=1}^{\infty} c_{ii'}p_{i'}\right)\left(\sum_{j'=1}^{\infty} c_{jj'}p_{j'}\right) \tag{42}$$

$$= \mathcal{P}_2 \sum_{i'=1}^{\infty}\sum_{j'=1}^{\infty} c_{ii'}c_{jj'}p_{i'}p_{j'} \tag{43}$$

$$= \sum_{i'=1}^{N/2}\sum_{j'=1}^{N/2} c_{ii'}c_{jj'}p_{i'}p_{j'} \tag{44}$$

$$= \left(\sum_{i'=1}^{N/2} c_{ii'}p_{i'}\right)\left(\sum_{j'=1}^{N/2} c_{jj'}p_{j'}\right) \tag{45}$$

$$= (\mathcal{P}_1\phi_i)(\mathcal{P}_1\phi_j) \,. \tag{46}$$

So ultimately we get

$$\mathcal{P}_2 f = \sum_{i\leq j}^{L} v_{ij}(\mathcal{P}_1\phi_i)(\mathcal{P}_1\phi_j) \,. \tag{47}$$

$\square$

## B.2 Rank Lemma

The following lemma is a generalization of the the Rank Lemma 4.5, which we will use for both the one- and high-dimensional cases. Ultimately, for an inner product $\langle \cdot, \cdot \rangle$ with certain orthogonality properties, it allows us to pass from function error $\|f - g\|^2$ to Frobenius norm error $\|F - G\|_F^2$ for some induced matrices $F, G$.

**Lemma B.2.** *Consider a commutative algebra equipped with an inner product, and a set of elements $\{p_t\}_{t=1}^{T}$. Suppose the terms $p_{\{t,t'\}} = p_t p_{t'}$, indexed by sets $\{t, t'\}$, are pairwise orthogonal, and normalized such that*

$$\|p_t p_{t'}\|^2 \geq \begin{cases} 1 & t \neq t' \\ 2 & t = t' \end{cases}$$

*Consider the terms:*

$$\phi_l = \sum_{t=1}^{T} c_{lt}p_t \,,$$

$$f = \sum_{l\leq l'}^{L} \frac{v_{ll'}}{1 + \mathbb{1}_{l=l'}}\phi_l\phi_{l'} \,,$$

$$g = \sum_{t\leq t'}^{T} \frac{g_{tt'}}{1 + \mathbb{1}_{t=t'}}p_t p_{t'} \,.$$

*Then we have the bound*

$$\|f - g\|^2 \geq \frac{1}{2}\|C^T V C - G\|_F^2 \,, \tag{48}$$

*where $C_{lt} = c_{lt}, V_{ll'} = v_{ll'}, G_{tt'} = g_{tt'}$, where we define $V$ and $G$ to be symmetric.*

*Proof.* To begin, we calculate inner products for $t \neq t'$:

$$\left\langle f, \frac{p_{\{t,t'\}}}{\|p_{\{t,t'\}}\|} \right\rangle = \frac{1}{\|p_{\{t,t'\}}\|} \left\langle \sum_{l \leq l'} \sum_{t,t'=1}^{T} \frac{v_{ll'}}{1 + \mathbb{1}_{l=l'}} c_{lt} c_{l't'} p_t p_{t'}, p_t p_{t'} \right\rangle \tag{49}$$

$$= \|p_t p_{t'}\| \sum_{l \leq l'}^{L} \frac{v_{ll'}}{1 + \mathbb{1}_{l=l'}} (c_{lt} c_{l't'} + c_{lt'} c_{l't}) \tag{50}$$

$$= \|p_t p_{t'}\| \left( \sum_{l=l'}^{L} \frac{v_{ll}}{2} (c_{lt} c_{lt'} + c_{lt'} c_{lt}) + \sum_{l<l'}^{L} v_{ll'} (c_{lt} c_{l't'} + c_{lt'} c_{l't}) \right) \tag{51}$$

$$= \|p_t p_{t'}\| \left( \sum_{l=l'}^{L} v_{ll} c_{lt} c_{lt'} + \sum_{l<l'}^{L} v_{ll'} (c_{lt} c_{l't'} + c_{lt'} c_{l't}) \right) . \tag{52}$$

Defining $v_{ll'} = v_{l'l}$, we may reindex and write the second sum as:

$$\sum_{l<l'}^{L} v_{ll'} (c_{lt} c_{l't'} + c_{lt'} c_{l't}) = \sum_{l<l'}^{L} v_{ll'} c_{lt} c_{l't'} + \sum_{l<l'}^{L} v_{ll'} c_{lt'} c_{l't} \tag{53}$$

$$= \sum_{l<l'}^{L} v_{ll'} c_{lt} c_{l't'} + \sum_{l>l'}^{L} v_{ll'} c_{lt} c_{l't'} . \tag{54}$$

So putting this together we get

$$\left\langle f, \frac{p_{\{t,t'\}}}{\|p_{\{t,t'\}}\|} \right\rangle = \|p_t p_{t'}\| \left( \sum_{l,l'}^{L} v_{ll'} c_{lt} c_{l't'} \right) = \|p_t p_{t'}\| [C^T V C]_{t,t'} .$$

By a similar calculation we conclude:

$$\left\langle f, \frac{p_{\{t,t\}}}{\|p_{\{t,t\}}\|} \right\rangle = \frac{\|p_t p_t\|}{2} [C^T V C]_{t,t} .$$

For $g$, we can directly calculate:

$$\left\langle g, \frac{p_{\{t,t'\}}}{\|p_{\{t,t'\}}\|} \right\rangle = \|p_t p_{t'}\| [G]_{t,t'} \tag{55}$$

$$\left\langle g, \frac{p_{\{t,t\}}}{\|p_{\{t,t\}}\|} \right\rangle = \frac{\|p_t p_t\|}{2} [G]_{t,t} . \tag{56}$$

Finally, by Parseval's Theorem we calculate:

$$\|f - g\|^2 = \sum_t \left( \left\langle f, \frac{p_{\{t,t\}}}{\|p_{\{t,t\}}\|} \right\rangle - \left\langle g, \frac{p_{\{t,t\}}}{\|p_{\{t,t\}}\|} \right\rangle \right)^2 + \sum_{t<t'}^{T} \left( \left\langle f, \frac{p_{\{t,t'\}}}{\|p_{\{t,t'\}}\|} \right\rangle - \left\langle g, \frac{p_{\{t,t'\}}}{\|p_{\{t,t'\}}\|} \right\rangle \right)^2 \tag{57}$$

$$= \sum_t \left( \left\langle f, \frac{p_{\{t,t\}}}{\|p_{\{t,t\}}\|} \right\rangle - \left\langle g, \frac{p_{\{t,t\}}}{\|p_{\{t,t\}}\|} \right\rangle \right)^2 + \frac{1}{2} \sum_{t \neq t'}^{T} \left( \left\langle f, \frac{p_{\{t,t'\}}}{\|p_{\{t,t'\}}\|} \right\rangle - \left\langle g, \frac{p_{\{t,t'\}}}{\|p_{\{t,t'\}}\|} \right\rangle \right)^2 \tag{58}$$

$$= \sum_t^{T} \frac{\|p_{\{t,t\}}\|^2}{4} [C^T V C - G]_{t,t}^2 + \frac{1}{2} \sum_{t \neq t'}^{T} \|p_{\{t,t'\}}\|^2 \cdot [C^T V C - G]_{t,t'}^2 \tag{59}$$

$$\geq \frac{1}{2} \sum_t^{T} [C^T V C - G]_{t,t}^2 + \frac{1}{2} \sum_{t \neq t'}^{T} [C^T V C - G]_{t,t'}^2 , \tag{60}$$

where in the last line we use our assumption on the lower bound of $\|p_{\{t,t'\}}\|^2$ and $\|p_{\{t,t\}}\|^2$. Hence:

$$\|f - g\|^2 \geq \frac{1}{2} \|C^T V C - G\|_F^2 . \tag{61}$$

$\square$

### B.3 Proof of one-dimensional Lower Bound

**Theorem B.3.** *Let $D = 1$. Then using the Vandermonde $L_2$ inner product over symmetric polynomials*

$$\max_{\|g\|_V=1} \min_{f \in Sym_L} \|f - g\|_V^2 \geq 1 - \frac{2L}{N} . \tag{62}$$

*In particular, for $L = \frac{N}{4}$ we recover a constant lower bound.*

*Proof.* We first build our counterexample $g$ by choosing its coefficients in the powersum basis, say:

$$g = \frac{1}{\sqrt{N}} \sum_{t=1}^{N/2} p_t p_t . \tag{63}$$

From orthogonality and the fact that $\|p_t p_t\|_V^2 = 2$ it's clear that $\|g\|_V = 1$, and note that $\mathcal{P}_2 g = g$. Applying Lemma B.1, for any $f \in \mathrm{Sym}_L$ we can write $\mathcal{P}_2 f$ in the form

$$\mathcal{P}_2 f = \sum_{i \leq j}^{L} v_{ij} (\mathcal{P}_1 \phi_i)(\mathcal{P}_1 \phi_j) . \tag{64}$$

One may also confirm that the Vandermonde inner product satisfies the requirements of Lemma B.2 when restricted to the range of $\mathcal{P}_2$, owing to the orthogonality property and the fact that for $1 \leq t, t' \leq N/2$:

$$\langle p_t p_{t'}, p_t p_{t'} \rangle_V = \begin{cases} 1 & t \neq t' \\ 2 & t = t' \end{cases}$$

So we've met all the necessary requirements to apply Lemma B.2 to $\mathcal{P}_2 f$ and $\mathcal{P}_2 g$, thus we have:

$$\min_{f \in \mathrm{Sym}_L} \|f - g\|_V^2 \geq \min_{f \in \mathrm{Sym}_L} \|\mathcal{P}_2 f - \mathcal{P}_2 g\|_V^2 \tag{65}$$

$$\geq \min_{C,V} \frac{1}{2} \|C^T V C - 2 * \frac{1}{\sqrt{N}} I\|_F^2 \tag{66}$$

$$= \min_{C,V} \frac{1}{N/2} \|C^T V C - I\|_F^2 , \tag{67}$$

where the factor of 2 appears based on the definition of the matrix $G$ in Lemma B.2

Note that $CVC^T \in \mathbb{C}^{N/2 \times N/2}$, but $V \in \mathbb{C}^{L \times L}$. So if $N/2 > L$, then $CVC^T$ is a rank-deficient approximation of the identity, and clearly we have

$$\min_{f \in \mathrm{Sym}_L} \|f - g\|_V^2 \geq \frac{N/2 - L}{N/2} = 1 - \frac{2L}{N} . \tag{68}$$

$\square$

## C  Exact statement of Main Result

### C.1  Theorem Statement

We begin by restating the main result, where for convenience we will change from $N$ set elements to $2N$.

We introduce the notation $\hat{D} := \min\left(D, \lfloor \sqrt{N/2} \rfloor\right)$. We also introduce the $L_2$ inner product

$$\langle f, g \rangle_{\mathcal{A}} = \mathbb{E}_{y \sim V; q, r \sim (S^1)^D} \left[ f(X(y, q, r)) \overline{g(X(y, q, r))} \right] , \tag{69}$$

where the set input $X(y, q, r) \in \mathbb{C}^{D \times 2N}$ with matrix entries $x_{dn}(y, q, r)$ is defined by:

$$x_{dn}(y, q, r) = \begin{cases} q_d y_n & 1 \leq n \leq N, \\ r_d y_{n-N} & N+1 \leq n \leq 2N. \end{cases} \tag{70}$$

And we restate the activation assumption in this new notation:

**Assumption C.1.** *The activation* $\sigma : \mathbb{C} \to \mathbb{C}$ *is analytic, and for a fixed* $D, N$ *there exist two-layer neural networks* $f_1, f_2$ *using* $\sigma$*, both with* $O\left(D^2 + D \log \frac{D}{\epsilon}\right)$ *width and* $O(D \log D)$ *bounded weights, such that:*

$$\sup_{|\xi| \leq 3} |f_1(\xi) - \xi^2| \leq \epsilon, \qquad \sup_{|\xi| \leq 3} \left| f_2(\xi) - \left(1 - (\xi/4)^{\min(D, \sqrt{N/2})}\right) \frac{\xi - 1/4}{\xi/4 - 1} \right| \leq \epsilon \tag{71}$$

Then our main theorem is thusly:

**Theorem C.2** (Exponential width-separation). *Fix* $2N$ *and* $D$ *such that* $\hat{D} > 1$*, and consider set elements* $X \in \mathbb{C}^{D \times 2N}$*. Define*

$$g(X) = -\frac{4N^2}{4\hat{D}} + \sum_{n, n'=1}^{2N} \prod_{d=1}^{\hat{D}} \left(1 - (x_{dn} x_{dn'}/4)^{\hat{D}}\right) \frac{x_{dn} x_{dn'} - 1/4}{x_{dn} x_{dn'}/4 - 1} \tag{72}$$

$$\tag{73}$$

*and* $g' = \frac{g}{\|g\|_{\mathcal{A}}}$*. Then the following is true:*

- *For* $L \leq N^{-2} \exp(O(\hat{D}))$*,*

$$\min_{f \in Sym_L} \|f - g'\|_{\mathcal{A}}^2 \geq \frac{1}{12}. \tag{74}$$

- *For* $L = 1$*, there exists* $f \in Sym_L^2$*, parameterized with an activation* $\sigma$ *that satisfies Assumption C.1, with width* $poly(N, D, 1/\epsilon)$*, depth* $O(\log D)$*, and maximum weight magnitude* $O(D \log D)$ *such that over the unit torus:*

$$\|f - g'\|_\infty \leq \epsilon. \tag{75}$$

**Remark 3.** *Let us remark about one aspect that will ease exposition. In the sequel, we will assume* $D \leq \sqrt{N/2}$ *so that* $\hat{D} = D$*. This is not a necessary assumption; in the case that* $D > \sqrt{N/2}$*, we can simply replace all instances of* $D$ *with* $\hat{D}$ *in the definition of* $g$ *and the subsequent proof. Because the data distribution has each row of* $X \in \mathbb{C}^{D \times 2N}$ *is i.i.d., the proof goes through exactly. Indeed, it would be equivalent to truncating each set vector to the first* $\hat{D}$ *elements. This will only impact the bounds by replacing* $D$ *with* $\hat{D}$*, in which circumstances we will clearly state.*

## C.2 Proof Roadmap

Let us roadmap the general proof.

In Section D.1, we justify the inner product $\langle \cdot, \cdot \rangle_{\mathcal{A}}$ and show it can be used to prove a high-dimensional analogue of the Projection Lemma (see Lemma D.2). In Section D.2 we further introduce a second inner product, whose orthogonality properties (see Theorem D.3) allow us to apply the Rank Lemma B.2. In Section D.3, we combine these results to first prove a lower bound for a simple choice of hard function (see Theorem D.4). Because this simple choice is not suitable for demonstrating the upper bound, we then conclude by showing the hard function $g'$ also evinces a lower bound via a similar argument (see Theorem D.5).

In Section E, we demonstrate the properties of the hard function $g$, by constructing the pieces of $g$ one by one and controlling their behavior, leading to Lemma E.3 which yields all the properties we need about $g$ for the rest of the proof.

In Section F we complete the proof of the upper bound. Specifically, in Section F.1 we show how to write $g'$ exactly in an analogous form to $\mathrm{Sym}_L^2$, but using very specific activations. In Section F.2 we write an approximation of this network in $\mathrm{Sym}_L^2$ using a given activation, and in Section F.3 we control the error between these two networks.

# D  Lower Bound of Main Result

## D.1  An $L_2$ inner product

As discussed in Section 5.2, we must first define an appropriate $L_2$ inner product, before we can prove a lower bound on function approximation. To that end, we will define an input distribution for the set inputs $X$.

Let us introduce several random variables: let $y \sim V$ as in the definition of the inner product $\langle \cdot, \cdot \rangle_V$ over $N$ variables. Let $q$ and $r$ be two random vectors of dimension $D$, with each entry $i.i.d.$ uniform on $S^1$.

Then we can define an input distribution for $X \in \mathbb{C}^{D \times 2N}$ with matrix entries $x_{dn}$:

$$x_{dn} = \begin{cases} q_d y_n & 1 \le n \le N \\ r_d y_{n-N} & N+1 \le n \le 2N \ . \end{cases} \tag{76}$$

The point of this assignment is how it transforms multisymmetric power sums:

$$\mathbf{p}_\alpha(X) = \frac{1}{\sqrt{|\alpha|}} \sum_{n=1}^{2N} \prod_d x_{dn}^{\alpha_d} \tag{77}$$

$$= \frac{1}{\sqrt{|\alpha|}} \sum_{n=1}^{N} \prod_d y_n^{\alpha_d} q_d^{\alpha_d} + \frac{1}{\sqrt{|\alpha|}} \sum_{n=1}^{N} \prod_d y_n^{\alpha_d} r_d^{\alpha_d} \tag{78}$$

$$= p_{|\alpha|}(y) \cdot (q^\alpha + r^\alpha) \ . \tag{79}$$

Then as stated before we have the inner product:

$$\langle f, g \rangle_{\mathcal{A}} = \mathbb{E}_{y \sim V, q \sim (S^1)^D, r \sim (S^1)^D} \left[ f(X) \overline{g(X)} \right] \ . \tag{80}$$

From our choices above we may use separability to write $\langle \cdot, \cdot \rangle_{\mathcal{A}}$ in terms of previously introduced inner products. For example:

$$\langle \mathbf{p}_\alpha, \mathbf{p}_\beta \rangle_{\mathcal{A}} = \mathbb{E}_{y,q,r} \left[ p_{|\alpha|}(y)(q^\alpha + r^\alpha) \overline{p_{|\beta|}(y)(q^\beta + r^\beta)} \right] \tag{81}$$

$$= \mathbb{E}_y \left[ p_{|\alpha|}(y) \overline{p_{|\beta|}(y)} \right] \mathbb{E}_{q,r} \left[ (q^\alpha + r^\alpha) \overline{(q^\beta + r^\beta)} \right] \tag{82}$$

$$= \langle p_{|\alpha|}, p_{|\beta|} \rangle_V \cdot \langle q^\alpha + r^\alpha, q^\beta + r^\beta \rangle_{S^1} \ . \tag{83}$$

We can now observe this inner product grants a "partial" orthogonality:

**Lemma D.1.** *Consider $\alpha, \beta \in \mathbb{N}^D$ with $1 \le |\alpha|, |\beta| \le N/2$. Then for $\gamma_k \in \mathbb{N}^D \setminus \{0\}$, if $K \ne 2$*

$$\left\langle \mathbf{p}_\alpha \mathbf{p}_\beta, \prod_{k=1}^{K} \mathbf{p}_{\gamma_k} \right\rangle_{\mathcal{A}} = 0 \ . \tag{84}$$

*Otherwise, for $K = 2$, we have:*

$$\langle \mathbf{p}_\alpha \mathbf{p}_\beta, \mathbf{p}_\gamma \mathbf{p}_\delta \rangle_{\mathcal{A}} = 2 \cdot (1 + \mathbb{1}_{|\alpha|=|\beta|}) \cdot \mathbb{1}_{\{|\alpha|,|\beta|\}=\{|\gamma|,|\delta|\}} \cdot (\mathbb{1}_{\alpha+\beta=\gamma+\delta} + \mathbb{1}_{(\alpha,\beta)=(\gamma,\delta)} + \mathbb{1}_{(\alpha,\beta)=(\delta,\gamma)}) \cdot \tag{85}$$

*Proof.* By separability, we can confirm that

$$\langle \mathbf{p}_\alpha \mathbf{p}_\beta, \prod_{k=1}^K \mathbf{p}_{\gamma_k} \rangle_{\mathcal{A}} = \langle p_{|\alpha|} p_{|\beta|}, \prod_{k=1}^K p_{|\gamma_k|} \rangle_V \cdot C \;, \tag{86}$$

where $C$ is the value of the expectation on the random variables $q$ and $r$. Thus if $K \neq 2$, because $|\alpha| + |\beta| \leq N$, this term is 0 by orthogonality of the Vandermonde inner product.

For the $K = 2$ case, we begin again by using separability:

$$\langle \mathbf{p}_\alpha \mathbf{p}_\beta, \mathbf{p}_\gamma \mathbf{p}_\delta \rangle_{\mathcal{A}} = \langle p_{|\alpha|} p_{|\beta|}, p_{|\gamma|} p_{|\delta|} \rangle_V \cdot \langle (q^\alpha + r^\alpha)(q^\beta + r^\beta), (q^\gamma + r^\gamma)(q^\delta + r^\delta) \rangle_{S^1} \;. \tag{87}$$

Let's consider first the inner product of power sums. Plugging in the definition of the normalizing constant $z_\lambda$ gives:

$$\langle p_{|\alpha|} p_{|\beta|}, p_{|\gamma|} p_{|\delta|} \rangle_V = (1 + \mathbb{1}_{|\alpha|=|\beta|}) \cdot \mathbb{1}_{\{|\alpha|,|\beta|\}=\{|\gamma|,|\delta|\}} \;.$$

Consider now the second inner product term. Noting that each element $q_d, r_d$ is i.i.d. uniform on the unit circle, orthogonality of the Fourier basis implies we can calculate this inner product by only including terms with matching exponents. Bearing in mind that $\alpha, \beta, \gamma, \delta \neq 0$, we must always have terms of the form $\langle q^{\alpha+\beta}, q^\gamma r^\delta \rangle_{S^1} = 0$, and therefore we distribute and calculate:

$$\langle q^{\alpha+\beta} + q^\alpha r^\beta + q^\beta r^\alpha + r^{\alpha+\beta}, q^{\gamma+\delta} + q^\gamma r^\delta + q^\delta r^\gamma + r^{\gamma+\delta} \rangle_{S^1}$$
$$= \langle q^{\alpha+\beta}, q^{\gamma+\delta} \rangle_{S^1} + \langle q^\alpha r^\beta + q^\beta r^\alpha, q^\gamma r^\delta + q^\delta r^\gamma \rangle_{S^1} + \langle r^{\alpha+\beta}, r^{\gamma+\delta} \rangle_{S^1}$$
$$= 2 \cdot \mathbb{1}_{\alpha+\beta=\gamma+\delta} + 2 \cdot \mathbb{1}_{(\alpha,\beta)=(\gamma,\delta)} + 2 \cdot \mathbb{1}_{(\alpha,\beta)=(\delta,\gamma)} \;.$$

Collecting the terms of both products and evaluating the indicator functions under all cases gives the result. $\qquad\square$

Looking at Equation 85, we can see inner product $\langle \cdot, \cdot \rangle_{\mathcal{A}}$ does not grant full orthogonality. The inner product gives orthogonality between powersum products of different lengths, but $\langle \mathbf{p}_\alpha \mathbf{p}_\beta, \mathbf{p}_\gamma \mathbf{p}_\delta \rangle_{\mathcal{A}}$ can be non-zero if $\alpha + \beta = \gamma + \delta$, even in the cases where $\{\alpha, \beta\} \neq \{\gamma, \delta\}$.

Nevertheless, this inner product still suffices to prove a similar result about projection for the $D > 1$ case.

Let $\mathcal{P}_1$ be the orthogonal projection onto $span(\{\mathbf{p}_\alpha : 1 \leq |\alpha|, |\beta| \leq N/2\})$ and $\mathcal{P}_2$ be the orthogonal projection onto $span(\{\mathbf{p}_\alpha \mathbf{p}_\beta : 1 \leq |\alpha|, |\beta| \leq N/2\})$. Here by orthogonal, we mean with respect to $\langle \cdot, \cdot \rangle_{\mathcal{A}}$.

**Lemma D.2.** *Given any $f \in Sym_L$ with $D > 1$, we may choose coefficients $v_{ij}$ over $i \leq j \leq L$, and multisymmetric polynomials $\phi_i$ over $i \leq L$, such that:*

$$\mathcal{P}_2 f = \sum_{i \leq j}^L v_{ij} (\mathcal{P}_1 \phi_i)(\mathcal{P}_2 \phi_j) \;. \tag{88}$$

*Proof.* As in Lemma B.1, if we approximate $\psi_l(x_n) = c_{l0} + \sum_{\alpha \neq 0} \frac{c_{l\alpha}}{\sqrt{|\alpha|}} x_n^\alpha$, then symmetrizing gives $\phi_l(X) = N c_{l0} + \sum_{\alpha \neq 0} c_{l\alpha} \mathbf{p}_\alpha$.

By a similar approximation as in Lemma B.1 that allows us to subtract out constant terms, we write:

$$f = \sum_{\eta \in \mathbb{N}^L} v_\eta \phi^\eta \;, \tag{89}$$

$$\phi_l = \sum_{\alpha \neq 0} c_{l\alpha} \mathbf{p}_\alpha \;. \tag{90}$$

Note that by Lemma D.1, $\langle \mathbf{p}_\alpha \mathbf{p}_\beta, \phi^\eta \rangle_{\mathcal{A}} = 0$ unless $|\eta| = 2$. So similarly to before, we may rewrite

$$\mathcal{P}_2 f = \sum_{|\eta|=2} v_\eta \mathcal{P}_2 \phi^\eta .$$

Here we can simplify notation. Let $\{e_i\}_{i=1}^L$ denote the standard basis vectors in dimension $L$. Every $\eta \in \mathbb{N}^L$ with $|\eta| = 2$ can be written as $\eta = e_i + e_j$, so let $v_{ij} := v_{e_i + e_j}$. Then we can rewrite:

$$\mathcal{P}_2 f = \sum_{i \leq j} v_{ij} \mathcal{P}_2 \phi_i \phi_j .$$

Again, by Lemma D.1, we know $\mathcal{P}_2$ will annihilate any term of the form $\mathbf{p}_\gamma \mathbf{p}_\delta$ if it's not the case that $1 \leq |\gamma|, |\delta| \leq N/2$. One can see this by noting that, for $1 \leq |\alpha|, |\beta| \leq N/2$, then $\{|\alpha|, |\beta|\} \neq \{|\gamma|, |\delta|\}$, and by the Lemma, $\langle \mathbf{p}_\alpha \mathbf{p}_\beta, \mathbf{p}_\gamma \mathbf{p}_\delta \rangle_{\mathcal{A}} = 0$.

So we may pass from $\mathcal{P}_2$ to $\mathcal{P}_1$:

$$\mathcal{P}_2 \phi_i \phi_j = \mathcal{P}_2 \left( \sum_{\gamma \in \mathbb{N}^D} c_{i\gamma} \mathbf{p}_\gamma \right) \left( \sum_{\delta \in \mathbb{N}^D} c_{j\delta} \mathbf{p}_\delta \right) \tag{91}$$

$$= \mathcal{P}_2 \sum_{\gamma \in \mathbb{N}^D} \sum_{\delta \in \mathbb{N}^D} c_{i\gamma} c_{j\delta} \mathbf{p}_\gamma \mathbf{p}_\delta \tag{92}$$

$$= \sum_{1 \leq |\gamma| \leq N/2} \sum_{1 \leq |\delta| \leq N/2} c_{i\gamma} c_{j\delta} \mathbf{p}_\gamma \mathbf{p}_\delta \tag{93}$$

$$= \left( \sum_{1 \leq |\gamma| \leq N/2} c_{i\gamma} \mathbf{p}_\gamma \right) \left( \sum_{1 \leq |\delta| \leq N/2} c_{j\delta} \mathbf{p}_\delta \right) \tag{94}$$

$$= (\mathcal{P}_1 \phi_i)(\mathcal{P}_1 \phi_j) . \tag{95}$$

So ultimately we get

$$\mathcal{P}_2 f = \sum_{i \leq j}^L v_{ij} (\mathcal{P}_1 \phi_i)(\mathcal{P}_1 \phi_j) . \tag{96}$$

$\square$

## D.2 A Diagonal Inner Product

Before we can apply Lemma B.2, which lets us transform function approximation error into matrix approximation error, we need a better inner product, one that is diagonal in the low-degree multisymmetric powersum basis.

Consider two more inner products, defined for $f, g$ in the range of $\mathcal{P}_2$:

$$\langle f, g \rangle_{\mathcal{A}_0} = \mathbb{E}_{y \sim V, q \sim (S^1)^D, r=0} \left[ f(X) \overline{g(X)} \right] . \tag{97}$$

This is nearly the same distribution as before, except we fix $r = 0$.

Then define

$$\langle f, g \rangle_* = \langle f, g \rangle_{\mathcal{A}} - 2 \langle f, g \rangle_{\mathcal{A}_0} . \tag{98}$$

Because $f$ and $g$ are restricted to the range of $\mathcal{P}_2$, we demonstrate positive-definiteness of this object, and therefore it is a valid inner product.

**Theorem D.3.** *The bilinear form* $\langle \cdot, \cdot \rangle_*$ *is an inner product when restricted to the range of* $\mathcal{P}_2$. *Furthermore, it is diagonal in the powersum basis* $p_{\{\alpha,\beta\}}$ *for* $1 \le |\alpha|, |\beta| \le N/2$.

*Proof.* Given $\mathbf{p}_\alpha \mathbf{p}_\beta, \mathbf{p}_\gamma \mathbf{p}_\delta \in im(\mathcal{P}_2)$, we can consider $\langle \mathbf{p}_\alpha \mathbf{p}_\beta, \mathbf{p}_\gamma \mathbf{p}_\delta \rangle_{\mathcal{A}_0}$ which can similarly be calculated via separability:

$$
\begin{aligned}
\langle \mathbf{p}_\alpha \mathbf{p}_\beta, \mathbf{p}_\gamma \mathbf{p}_\delta \rangle_{\mathcal{A}_0} &= \langle p_{|\alpha|} p_{|\beta|}, p_{|\gamma|} p_{|\delta|} \rangle_V \cdot \langle q^{\alpha+\beta}, q^{\gamma+\delta} \rangle_{S^1} \\
&= (1 + \mathbb{1}_{|\alpha|=|\beta|}) \cdot \mathbb{1}_{\{|\alpha|,|\beta|\}=\{|\gamma|,|\delta|\}} \cdot \mathbb{1}_{\alpha+\beta=\gamma+\delta} .
\end{aligned}
$$

It follows from Lemma D.1 that:

$$
\begin{aligned}
\langle \mathbf{p}_\alpha \mathbf{p}_\beta, \mathbf{p}_\gamma \mathbf{p}_\delta \rangle_* &= \langle \mathbf{p}_\alpha \mathbf{p}_\beta, \mathbf{p}_\gamma \mathbf{p}_\delta \rangle_{\mathcal{A}} - 2 \langle \mathbf{p}_\alpha \mathbf{p}_\beta, \mathbf{p}_\gamma \mathbf{p}_\delta \rangle_{\mathcal{A}_0} \\
&= 2 \cdot (1 + \mathbb{1}_{|\alpha|=|\beta|}) \cdot (\mathbb{1}_{(\alpha,\beta)=(\gamma,\delta)} + \mathbb{1}_{(\alpha,\beta)=(\delta,\gamma)}) .
\end{aligned}
$$

To eliminate the ambiguity of $\mathbf{p}_\alpha \mathbf{p}_\beta$ vs. $\mathbf{p}_\beta \mathbf{p}_\alpha$, let us define $\mathbf{p}_{\{\alpha,\beta\}}$ equal to both these terms. Then we can equivalently write:

$$
\langle \mathbf{p}_{\{\alpha,\beta\}}, \mathbf{p}_{\{\gamma,\delta\}} \rangle_* = 2 \cdot (1 + \mathbb{1}_{|\alpha|=|\beta|}) \cdot (1 + \mathbb{1}_{\alpha=\beta}) \cdot \mathbb{1}_{\{\alpha,\beta\}=\{\gamma,\delta\}} .
$$

Evaluating the indicator functions under all cases we can see:

$$
\langle \mathbf{p}_\alpha \mathbf{p}_\beta, \mathbf{p}_\gamma \mathbf{p}_\delta \rangle_* = \begin{cases} 0 & \{\alpha,\beta\} \neq \{\gamma,\delta\} \\ 2 & \{\alpha,\beta\} = \{\gamma,\delta\}, \quad |\alpha| \neq |\beta| \\ 4 & \{\alpha,\beta\} = \{\gamma,\delta\}, \quad |\alpha| = |\beta|, \quad \alpha \neq \beta \\ 8 & \{\alpha,\beta\} = \{\gamma,\delta\}, \quad \alpha = \beta \end{cases}
$$

Then we've shown that the bilinear form $\langle \cdot, \cdot \rangle_*$, treated as a matrix in the basis of all $\mathbf{p}_{\{\alpha,\beta\}}$, is positive-definite and diagonal. Since this basis spans the range of $\mathcal{P}_2$, it follows that the bilinear form is an inner product. $\qquad\square$

### D.3 Proof of Lower Bound

We first prove a lower bound using a slightly simpler hard function $g$, before updating the argument to the true choice of $g$ further below.

**Theorem D.4.** *Let $D > 1$. In particular, assume $\min(N/2, D-1) \ge 2$. Then we have*

$$
\max_{\|g\|_{\mathcal{A}}=1} \min_{f \in Sym_L} \|f - g\|_{\mathcal{A}}^2 \ge \frac{1}{6} - \frac{L}{6 \cdot 2^{\min(N/2, D-1)}} . \tag{99}
$$

*So for $L \le 2^{\min(N/2, D-1)-3}$ we have a constant lower bound on the approximation error.*

*Proof.* Define $T = |\{\alpha \in \mathbb{N}^D : |\alpha| = N/2\}|$ and choose the bad function $g = \frac{1}{\sqrt{12T}} \sum_{|\alpha|=N/2} \mathbf{P}_{\{\alpha,\alpha\}}$.

Observe that although $\langle \cdot, \cdot \rangle_{\mathcal{A}}$ is not fully orthogonal in the powersum basis, we can nevertheless calculate by Lemma D.1 that for $|\alpha| = |\beta| = N/2$:

$$
\begin{aligned}
\langle \mathbf{P}_{\{\alpha,\alpha\}}, \mathbf{P}_{\{\beta,\beta\}} \rangle_{\mathcal{A}} &= 4 \cdot (\mathbb{1}_{\alpha+\alpha=\beta+\beta} + \mathbb{1}_{(\alpha,\alpha)=(\beta,\beta)} + \mathbb{1}_{(\alpha,\alpha)=(\beta,\beta)}) && (100) \\
&= 12 \cdot \mathbb{1}_{\alpha=\beta} . && (101)
\end{aligned}
$$

Therefore we can confirm that $g$ is normalized:

$$\|g\|_{\mathcal{A}}^2 = \frac{1}{12T} \sum_{|\alpha|=N/2} \sum_{|\beta|=N/2} \langle \mathbf{P}_{\{\alpha,\alpha\}}, \mathbf{P}_{\{\beta,\beta\}} \rangle_{\mathcal{A}} \tag{102}$$

$$= \frac{1}{12T} \sum_{|\alpha|=N/2} \sum_{|\alpha|=N/2} 12 \cdot \mathbb{1}_{\alpha=\beta} \tag{103}$$

$$= \frac{1}{T} \sum_{|\alpha|=N/2} 1 \tag{104}$$

$$= 1 . \tag{105}$$

Again, we have $\mathcal{P}_2 g = g$. Now by Lemma D.2, we may write:

$$\mathcal{P}_2 f = \sum_{i \leq j}^{L} v_{ij} (\mathcal{P}_1 \phi_i)(\mathcal{P}_1 \phi_j) .$$

Finally, note that $\langle \cdot, \cdot \rangle_*$ obeys the inner product conditions of Lemma B.2 on the range of $\mathcal{P}_2$, following from orthogonality and the normalization:

$$\langle \mathbf{P}_\alpha \mathbf{P}_\beta, \mathbf{P}_\alpha \mathbf{P}_\beta \rangle_* = \begin{cases} 2 & |\alpha| \neq |\beta| \\ 4 & |\alpha| = |\beta|, \quad \alpha \neq \beta \\ 8 & \alpha = \beta \end{cases}$$

So we can apply Lemma B.2 to $\mathcal{P}_2 f, \mathcal{P}_2 g$, and the inner product $\langle \cdot, \cdot \rangle_*$. Hence, we can derive:

$$\min_{f \in \mathrm{Sym}_L} \|f - g\|_{\mathcal{A}}^2 \overset{(a)}{\geq} \min_{f \in \mathrm{Sym}_L} \|\mathcal{P}_2 f - \mathcal{P}_2 g\|_{\mathcal{A}}^2 \tag{106}$$

$$\overset{(b)}{\geq} \min_{f \in \mathrm{Sym}_L} \|\mathcal{P}_2 f - \mathcal{P}_2 g\|_*^2 \tag{107}$$

$$\overset{(c)}{\geq} \min_{C,V} \frac{1}{2} \|C^T V C - 2 * \frac{1}{\sqrt{12T}} I\|_F^2 \tag{108}$$

$$= \min_{C,V} \frac{1}{6T} \|C^T V C - I\|_F^2 . \tag{109}$$

Here, $(a)$ follows from the definition of $\mathcal{P}_2$ as an orthogonal projection with respect to $\langle \cdot, \cdot \rangle_{\mathcal{A}}$, $(b)$ follows from the fact that $\|\cdot\|_{\mathcal{A}}^2 \geq \|\cdot\|_*^2$, and $(c)$ follows from the application of Lemma B.2.

These matrices are elements of $\mathbb{C}^{T \times T}$, but the term $C^T V C$ is constrained to rank $L$. Hence, as before we calculate:

$$\min_{f \in \mathrm{Sym}_L} \|f - g\|_{\mathcal{A}}^2 \geq \frac{T - L}{6T} = \frac{1}{6} - \frac{L}{6T} . \tag{110}$$

Letting $m = \min(N/2, D - 1)$ and assuming $m \geq 2$, it is a simple bound to calculate

$$T = \binom{N/2 + D - 1}{N/2} \geq \binom{2m}{m} \approx \frac{4^m}{\sqrt{\pi m}} \geq 2^m ,$$

and the bound follows.

$\square$

This theorem demonstrates a hard function $g$ that cannot be efficiently approximated by $f \in \mathrm{Sym}_L$ for $L = poly(N, D)$, but it does not yet evince a separation. Indeed, observing that $\|g\|_\infty =$

$\frac{1}{\sqrt{12T}}N^2T = \frac{N^2\sqrt{T}}{\sqrt{12}}$, $g$ has very large magnitude, and there's no obvious way to easily approximate this function by an efficient network in $\mathrm{Sym}_L^2$.

Thus, we consider a more complicated choice for $g$, that allows for the separation:

**Theorem D.5.** *Let $D > 1$. Then let $g' = \frac{g}{\|g\|_{\mathcal{A}}}$ for $g$ as defined in Lemma E.3, such that $\|g'\|_{\mathcal{A}} = 1$. Then for $L \leq N^{-2}\exp(O(D))$:*

$$\min_{f \in Sym_L} \|f - g'\|_{\mathcal{A}}^2 \geq \frac{1}{12} . \tag{111}$$

*Proof.* The lower bound follows almost identically as before. By Lemma E.3.4 we still have that $\mathcal{P}_2 g' = g'$. So we can write

$$g = \sum_{1 \leq |\alpha| \leq N/2} g_\alpha \mathbf{P}_{\{\alpha,\alpha\}} \tag{112}$$

$$g' = \sum_{1 \leq |\alpha| \leq N/2} \frac{g_\alpha}{\|g\|_{\mathcal{A}}} \mathbf{P}_{\{\alpha,\alpha\}} . \tag{113}$$

Thus, by the same reasoning as Theorem D.4 we recover the lower bound:

$$\min_{f \in \mathrm{Sym}_L} \|f - g'\|_{\mathcal{A}}^2 \geq \min_{f \in \mathrm{Sym}_L} \|\mathcal{P}_2 f - \mathcal{P}_2 g'\|_{\mathcal{A}}^2 \tag{114}$$

$$\geq \min_{f \in \mathrm{Sym}_L} \|\mathcal{P}_2 f - \mathcal{P}_2 g'\|_*^2 \tag{115}$$

$$\geq \min_{C,V} \frac{1}{2} \|C^T V C - G'\|_F^2 , \tag{116}$$

where $G'$ is the matrix induced by $g'$ as given in Lemma B.2, i.e. the diagonal matrix indexed by $G'_{\alpha\alpha} = \frac{2g_\alpha}{\|g\|_{\mathcal{A}}}$.

Now, by the partial orthogonality of $\langle \cdot, \cdot \rangle_{\mathcal{A}}$ noted in Lemma D.1, we have:

$$\|g\|_{\mathcal{A}}^2 = \sum_{1 \leq |\alpha| \leq N/2} \sum_{1 \leq |\beta| \leq N/2} \langle g_\alpha \mathbf{P}_{\{\alpha,\alpha\}}, g_\beta \mathbf{P}_{\{\beta,\beta\}} \rangle_{\mathcal{A}} \tag{117}$$

$$= \sum_{1 \leq |\alpha| \leq N/2} \sum_{1 \leq |\beta| \leq N/2} g_\alpha \overline{g_\beta} (12 \cdot \mathbb{1}_{\alpha=\beta}) \tag{118}$$

$$= 12 \sum_{1 \leq |\alpha| \leq N/2} |g_\alpha|^2 . \tag{119}$$

Hence, we can say

$$\|G'\|_F^2 = \sum_{1 \leq |\alpha| \leq N/2} \left| \frac{2g_\alpha}{\|g\|_{\mathcal{A}}} \right|^2 \tag{120}$$

$$= \frac{4 \sum_{1 \leq |\alpha| \leq N/2} |g_\alpha|^2}{12 \sum_{1 \leq |\alpha| \leq N/2} |g_\alpha|^2} \tag{121}$$

$$= \frac{1}{3} . \tag{122}$$

Call $G'_L$ the best rank-$L$ approximation of $G'$ in the Frobenius norm. By classical properties of SVD it follows that $G'_L$ is a diagonal matrix with $L$ entries corresponding to the $L$ largest elements of $G'$. Then because $\|G'\|_F^2 = \frac{1}{3}$:

$$\|G'_L - G'\|_F^2 = \frac{1}{3} - \sum_{l=1}^{L} \left( \frac{|2g_{\alpha_l}|}{\|g\|_{\mathcal{A}}} \right)^2 , \tag{123}$$

where we order $|g_{\alpha_l}|$ in non-increasing order.

Combining Lemma E.3.2 and E.3.4 yields the inequality that for all $\alpha$ such that $1 \leq |\alpha| \leq N/2$:

$$\left(\frac{|2g_\alpha|}{\|g\|_{\mathcal{A}}}\right)^2 \leq 4N^2 \left(1 - \left(\frac{1}{4}\right)^2\right)^{2D}, \tag{124}$$

so we can conclude

$$\min_{f \in \mathrm{Sym}_L} \|f - g'\|_{\mathcal{A}}^2 \geq \frac{1}{2}\|G'_L - G'\|_F^2 \tag{125}$$

$$\geq \frac{1}{6} - 2LN^2 \left(1 - \left(\frac{1}{4}\right)^2\right)^{2D}. \tag{126}$$

Hence, if $L \leq \frac{1}{24} \cdot N^{-2} \left(\frac{16}{15}\right)^{2D}$, we derive a lower bound:

$$\min_{f \in \mathrm{Sym}_L} \|f - g'\|_{\mathcal{A}}^2 \geq \frac{1}{12}. \tag{127}$$

$\square$

We remark here that in the instance $D > \sqrt{N/2}$, we replace $D$ with $\hat{D}$ in the above bound, which is consistent with Theorem C.2.

# E   Definition of hard function $g$

In this section we incrementally build the (unnormalized) hard function $g$, ultimately for the sake of Lemma E.3. This lemma characterizes all the properties of $g$ that we need to guarantee the lower and upper bounds.

**Remark 4.** *In the following section, we assume $D \leq \sqrt{N/2}$ for simplicity of exposition. In the case that $D > \sqrt{N/2}$, we replace all instances of $D$ in our functional definitions with $\hat{D} = \min(D, \sqrt{N/2})$, which is only necessary for a projection argument in Lemma E.3 and makes no meaningful change to the proofs.*

## E.1   Mobius transform

We begin with the following, with $\xi \in \mathbb{C}$ and $|\xi| = 1$. And in the sequel, we always fix $r = 1/4$. Consider the 1-D Mobius transformation, with its truncated variant with $t \geq 1$:

$$\mu(\xi) = \frac{\xi - r}{r\xi - 1} \tag{128}$$

$$\hat{\mu}_t(\xi) = \left(1 - (r\xi)^t\right) \cdot \mu(\xi) \tag{129}$$

$$= (r - \xi) \cdot \left(1 + r\xi + (r\xi)^2 + \cdots + (r\xi)^{t-1}\right) \tag{130}$$

**Lemma E.1.** *The following properties hold (where infinity norms are defined with respect to $S^1$):*

1. $\|\mu\|_\infty = 1$

2. $\|\mu\|_{S^1} = 1$

3. $\|\hat{\mu}_t\|_\infty \leq 1 + r^t$

4. $\|\hat{\mu}_t\|_{S^1}^2 = 1 + r^{2t}$

5. $\langle \hat{\mu}_t, 1 \rangle_{S^1} = r$, $\langle \hat{\mu}_t, \xi \rangle_{S^1} = r^2 - 1$ and $|\langle \hat{\mu}_t, \xi^a \rangle_{S^1}| < 1 - r^2$ for all $a \geq 2$

6. For $|\xi| = 1, |\omega| \leq 1 + \frac{1}{t}$,

$$|\hat{\mu}_t(\xi) - \hat{\mu}_t(\omega)| \leq 6|\xi - \omega| \tag{131}$$

*Proof.* It is a fact [8] that $\mu$ analytically maps the unit disk to itself, and additional the unit circle to itself, i.e. for any $|\xi| = 1$ we have $|\mu(\xi)| = 1$. Hence $\|\mu\|_\infty = \|\mu\|_{S^1} = 1$.

We can see that truncation gently perturbs this fact, so for $|\xi| = 1$:

$$|\hat{\mu}_t(\xi)| = |1 - (r\xi)^t| \cdot |\mu(\xi)| \tag{132}$$
$$\leq 1 + r^t \tag{133}$$

Additionally, we can calculate the coefficient on each monomial in $\hat{\mu}$:

$$\langle \hat{\mu}_t, \xi^a \rangle_{S^1} = \begin{cases} r & a = 0 \\ -(r^{a-1} - r^{a+1}) & 1 \leq a \leq t - 1 \\ -r^{t-1} & a = t \\ 0 & a \geq t \end{cases} \tag{134}$$

It is easy to confirm that the value of $|\langle \hat{\mu}_t, \xi^a \rangle_{S^1}|$ is maximized at $a = 1$. Hence, we can write the $L_2$ norm:

$$\|\hat{\mu}_t\|_{S^1}^2 = \sum_{a=0}^\infty |\langle \hat{\mu}_t, \xi^a \rangle_{S^1}|^2 \tag{135}$$

$$= r^2 + \sum_{a=1}^{t-1} (r^{a-1} - r^{a+1})^2 + r^{2t-2} \tag{136}$$

$$= r^2 + \sum_{a=1}^{t-1} (r^{2a-2} - 2r^{2a} + r^{2a+2}) + r^{2t-2} \tag{137}$$

$$= 1 + r^{2t} \tag{138}$$

Finally, for $|\xi| = 1, |\omega| \leq 1 + \frac{1}{t} \leq 2$:

$$|\mu(\xi) - \mu(\omega)| = \left| \frac{\xi - r}{r\xi - 1} - \frac{\omega - r}{r\omega - 1} \right| \tag{139}$$

$$= \left| \frac{(r^2 - 1)(\xi - \omega)}{(r\xi - 1)(r\omega - 1)} \right| . \tag{140}$$

So noting $r = \frac{1}{4}$ we get

$$|\mu(\xi) - \mu(\omega)| \leq \frac{8}{3}|\xi - \omega| . \tag{141}$$

Thus:

$$|\hat{\mu}(\xi) - \hat{\mu}(\omega)| = \left| (1 - (r\xi)^t) \cdot \mu(\xi) - (1 - (r\omega)^t) \cdot \mu(\omega) \right| \tag{142}$$

$$\leq \left| (1 - (r\xi)^t) \cdot \mu(\xi) - (1 - (r\omega)^t) \cdot \mu(\xi) \right| + \left| (1 - (r\omega)^t) \cdot \mu(\xi) - (1 - (r\omega)^t) \cdot \mu(\omega) \right| \tag{143}$$

$$\leq |\mu(\xi)| \cdot r^t |\xi^t - \omega^t| + |1 - (r\omega)^t| \cdot |\mu(\xi) - \mu(\omega)| \tag{144}$$

$$\leq r^t |\xi^t - \omega^t| + |1 - (r\omega)^t| \cdot \frac{8}{3}|\xi - \omega| . \tag{145}$$

Note that for $|\xi| = 1, |\omega| \leq 1 + \frac{1}{t}$, because $|\omega|^k \leq e$ for $k \leq t$, we have

$$\left| \xi^t - \omega^t \right| = \left| (\xi - \omega)(\xi^{t-1} + \xi^{t-2}\omega + \cdots + \xi\omega^{t-2} + \omega^{t-1} \right| \leq et|\xi - \omega| . \tag{146}$$

Further plugging in that $r = \frac{1}{4}$ and $t \geq 1$:

$$|\hat{\mu}(\xi) - \hat{\mu}(\omega)| \leq 4^{-t}et|\xi - \omega| + \left(1 + 4^{-t}e\right) \cdot \frac{8}{3}|\xi - \omega| \tag{147}$$

$$< 6|\xi - \omega| . \tag{148}$$

$\square$

## E.2  $h$ **function**

Now, consider $z \in \mathbb{C}^D$ with $|z_i| = 1$ for all $i$. We now define:

$$h(z) = \prod_{i=1}^{D} \hat{\mu}_D(z_i) . \tag{149}$$

**Lemma E.2.** *The following are true:*

1. $\|h\|_\infty \leq 1 + 2^{-D}$

2. $1 \leq \|h\|^2_{S^1} \leq 1 + 2^{-D}$

3. *For $z, z' \in (S^1)^D$*

$$|h(z) - h(z')| \leq 12\|z - z'\|_1 .$$

*Proof.* We can immediately bound:

$$\|h\|_\infty = \prod_{i=1}^{D} \|\hat{\mu}_D\|_\infty \tag{150}$$

$$\overset{(a)}{\leq} \left(1 + r^D\right)^D \tag{151}$$

$$\overset{(b)}{\leq} 1 + 2^D \cdot r^D \tag{152}$$

$$\leq 1 + 2^{-D} , \tag{153}$$

where $(a)$ follows from Lemma E.1.3 and $(b)$ follows from the binomial identity that $(1 + x)^t \leq 1 + 2^t x$ for $x \in [0, 1], t \geq 1$. In the last line we simply plug in $r = 1/4$.

Similarly by Lemma E.1.4,

$$\|h\|^2_{S^1} = \prod_{i=1}^{D} \|\hat{\mu}_D\|^2_{S^1} \tag{154}$$

$$= \left(1 + r^{2D}\right)^D \tag{155}$$

$$\leq \left(1 + r^D\right)^D . \tag{156}$$

And so by the same binomial inequality, we have

$$1 \leq \|h\|^2_{S^1} \leq 1 + 2^{-D} . \tag{157}$$

Finally, observe that:

$$|h(z) - h(z')| \leq \sum_{i=1}^{D} \left| \left( \prod_{j=1}^{i-1} \hat{\mu}_D(z_j) \right) (\hat{\mu}_D(z_i) - \hat{\mu}_D(z'_i)) \left( \prod_{j=i+1}^{D} \hat{\mu}_D(z'_j) \right) \right| \tag{158}$$

$$\overset{(a)}{\leq} \sum_{i=1}^{D} |\hat{\mu}_D(z_i) - \hat{\mu}_D(z'_i)| \, (1 + r^D)^{D-1} \tag{159}$$

$$\overset{(b)}{\leq} 6 \sum_{i=1}^{D} |z_i - z'_i| \left( 1 + 2^{-D} \right) \tag{160}$$

$$\leq 12 \|z - z'\|_1 \, , \tag{161}$$

where in $(a)$ we apply E.1.3, and in $(b)$ we apply E.1.6 and the same binomial identity as above. $\quad\square$

## E.3  $g$ **function**

Now, reminding $z_{n,n'} = x_n \circ x_{n'}$, let:

$$g(X) = -4N^2 r^D + \sum_{n,n'=1}^{2N} h(z_{n,n'}) \, . \tag{162}$$

Note that we subtract a constant here to ensure $g$ has no constant term, which will be necessary for the fact $\mathcal{P}_2 g = g$.

**Remark 5.** *The following lemma is the only place we explicitly require the assumption $D \leq \sqrt{N/2}$, as this guarantees that $\mathcal{P}_2 g = g$. In the case that $D > \sqrt{N/2}$, we simply replace all instances of $D$ in this section with $\hat{D} = \min(D, \sqrt{N/2})$. This ensures $g$ is only supported on $\mathbf{p}_{\{\alpha,\alpha\}}$ with $|\alpha| \leq \hat{D}^2 \leq N/2$. And the subsequent proofs are identical.*

**Lemma E.3.** *The following are true:*

1. $\|g\|_\infty \leq 12N^2$.

2. $1 \leq \|g\|_{\mathcal{A}}^2 \leq 3N^2(1 + 2^{-D})$.

3. $\mathcal{P}_2 g = g$.

4. *We may write* $g = \sum_{1 \leq |\alpha| \leq N/2} g_\alpha \mathbf{p}_{\{\alpha,\alpha\}}$, *where* $|g_\alpha|^2 \leq N^2(1 - r^2)^{2D}$.

5. $\mathrm{Lip}(g) \leq 48N\sqrt{ND}$.

*Proof.* First, it's easy to see from Lemma E.2.1

$$\|g\|_\infty \leq |-4N^2 r^D| + 4N^2 \|h\|_\infty \tag{163}$$

$$\leq 4N^2 \left( 2^{-2D} + 1 + 2^{-D} \right) \tag{164}$$

$$\leq 12N^2 \, . \tag{165}$$

Let us expand $h$ as

$$h(z) = \sum_{\|\alpha\|_\infty \leq D} h_\alpha z^\alpha \, , \tag{166}$$

noting that by definition of $\hat{\mu}_D$ and Lemma E.1.5 we have the constant term $h_0 = r^D$.

Now we can expand

$$g(X) = -4N^2 r^D + \sum_{n,n'=1}^{2N} h(z_{n,n'}) \tag{167}$$

$$= -4N^2 r^D + \sum_{n,n'=1}^{2N} \left[ r^D + \sum_{1 \leq \|\alpha\|_\infty \leq D} h_\alpha z_{n,n'}^\alpha \right] \tag{168}$$

$$= \sum_{n,n'=1}^{2N} \sum_{1 \leq \|\alpha\|_\infty \leq D} h_\alpha z_{n,n'}^\alpha \tag{169}$$

$$= \sum_{1 \leq \|\alpha\|_\infty \leq D} h_\alpha \sum_{n,n'=1}^{2N} \prod_{d=1}^{D} (x_{dn} x_{dn'})^{\alpha_d} \tag{170}$$

$$= \sum_{1 \leq \|\alpha\|_\infty \leq D} h_\alpha |\alpha| \left( \frac{1}{\sqrt{|\alpha|}} \sum_{n=1}^{2N} \prod_{d=1}^{D} x_{dn}^{\alpha_d} \right) \left( \frac{1}{\sqrt{|\alpha|}} \sum_{n'=1}^{2N} \prod_{d'=1}^{D} x_{d'n'}^{\alpha_{d'}} \right) \tag{171}$$

$$= \sum_{1 \leq \|\alpha\|_\infty \leq D} h_\alpha |\alpha| \mathbf{p}_{\{\alpha,\alpha\}}(X) . \tag{172}$$

Note that $\|\alpha\|_\infty \leq D$ implies $|\alpha| \leq D^2 \leq N/2$, so it clearly follows that $\mathcal{P}_2 g = g$. So by Lemma D.1, $\langle \mathbf{p}_{\{\alpha,\alpha\}}, \mathbf{p}_{\{\beta,\beta\}} \rangle_\mathcal{A} = 12 \cdot \mathbb{1}_{\alpha=\beta}$ whenever $1 \leq |\alpha|, |\beta| \leq N/2$, so we can handily calculate:

$$\|g\|_\mathcal{A}^2 = \sum_{1 \leq \|\alpha\|_\infty \leq D} h_\alpha^2 |\alpha|^2 \|\mathbf{p}_{\{\alpha,\alpha\}}\|_\mathcal{A}^2 \tag{173}$$

$$\leq 12 \cdot (N/2)^2 \sum_{1 \leq \|\alpha\|_\infty \leq D} h_\alpha^2 \tag{174}$$

$$\leq 3N^2 \|h\|_{S^1}^2 \tag{175}$$

$$\leq 3N^2 \left( 1 + 2^{-D} \right) , \tag{176}$$

where the last line uses Lemma E.2.2.

And likewise

$$\|g\|_\mathcal{A}^2 = \sum_{1 \leq \|\alpha\|_\infty \leq D} h_\alpha^2 |\alpha|^2 \|\mathbf{p}_{\{\alpha,\alpha\}}\|_\mathcal{A}^2 \tag{177}$$

$$\geq 12 \left( -r^D + \sum_{\|\alpha\|_\infty \leq D} h_\alpha^2 \right) \tag{178}$$

$$= 12(-r^D + \|h\|_{S^1}^2) \tag{179}$$

$$\geq 1 , \tag{180}$$

and the last line again uses Lemma E.2.2. Finally, note that for any $\alpha$ such that $|\alpha| \leq N/2$, applying Lemma E.1.5.

$$|g_\alpha|^2 = |h_\alpha |\alpha||^2 = |\alpha|^2 \prod_{i=1}^{D} |\langle \hat{\mu}_D, \xi^{\alpha_i} \rangle_{S^1}|^2 \tag{181}$$

$$\leq N^2 (1 - r^2)^{2D} . \tag{182}$$

Finally we consider the Lipschitz norm. For $X, \hat{X} \in \mathbb{C}^{D \times 2N}$ with each entry of unit norm, it's easy to confirm by Lemma E.2.3 that:

$$|g(X) - g(\hat{X})| \leq \sum_{n,n'=1}^{2N} |h(z_{n,n'}) - h(\hat{z}'_{n,n'})| \tag{183}$$

$$\leq 12 \sum_{n,n'=1}^{2N} \|z_{n,n'} - \hat{z}_{n,n'}\|_1 \tag{184}$$

$$= 12 \sum_{n,n'=1}^{2N} \sum_{d=1}^{D} |x_{dn} x_{dn'} - \hat{x}_{dn} \hat{x}_{dn'}| \tag{185}$$

$$\leq 12 \sum_{n,n'=1}^{2N} \sum_{d=1}^{D} |x_{dn}| \cdot |x_{dn'} - \hat{x}_{dn'}| + |\hat{x}_{dn'}| \cdot |x_{dn} - \hat{x}_{dn}| \tag{186}$$

$$= 48N \sum_{n=1}^{2N} \sum_{d=1}^{D} |x_{dn} - \hat{x}_{dn}| \tag{187}$$

$$= 48N \|X - \hat{X}\|_1 \tag{188}$$

$$\leq 48N \sqrt{2ND} \|X - \hat{X}\|_2 \tag{189}$$

$\square$

# F    Upper Bound of Main Result

In this section we prove the upper bound to representing $g$ with an admissible activation that satisfies Assumption C.1.

The strategy is as follows. In Section F.1 we exactly encode the hard function $g$ with an efficient network, but allowing the choice of very particular activation functions. In Section F.2, we leverage Assumption C.1 to build a network that approximates the exact one, using a given activation. We complete the proof in Section F.3 by showing the exact and approximate networks stay close together, inducting through the layers.

## F.1    Exact Representation

Let us first describe how to write $g$ exactly with a network in $\text{Sym}_L^2$, using particular activations. We can then demonstrate to approximate those activations, which only introduces a polynomial dependence in the desired error bound $\epsilon$.

For exact representation, the activations we will allow are $\xi \to \xi^2$, and $\xi \to \hat{\mu}_D(\xi)$. Note that from the fact that $\xi \cdot \omega = \frac{1}{2}\left((\xi + \omega)^2 - \xi^2 - \omega^2\right)$, we can exactly multiply scalars with these activations.

Then consider the following structure for $f \in \text{Sym}_L^2$ with $L = 1$. Given $x, x' \in \mathbb{C}^D$ with $|x_i| = |x'_i| = 1$ for all $i$, we define $\psi_1^*(x, x')$ via a network as follows. In particular, we will use $\cdot$ to explicitly indicate all scalar multiplication:

$$z^* = (x_1 \cdot x'_1, \ldots, x_D \cdot x'_D) \tag{190}$$

$$Z^{(1)*} = (\hat{\mu}_D(z_1^*), \ldots, \hat{\mu}_D(z_D^*)) \in \mathbb{C}^D \tag{191}$$

$$Z^{(2)*} = \left(Z_1^{(1)*} \cdot Z_2^{(1)*}, \ldots, Z_{D-1}^{(1)*} \cdot Z_D^{(1)*}\right) \in \mathbb{C}^{D/2} \tag{192}$$

$$\cdots \tag{193}$$

$$Z^{(\log_2 D)*} = Z_1^{(\log_2 D-1)*} \cdot Z_2^{(\log_2 D-1)*} \in \mathbb{C} \tag{194}$$

$$\psi_1^*(x, x') = Z^{(\log_2 D)*} \tag{195}$$

In other words, we exactly calculate $\psi_1^*(x, x') = h(x \circ x')$ through $\log_2 D$ layers by multiplying the terms $\hat{\mu}_D(z_i)$ at each layer. Note that $|z_i^*| = 1$ for all $i$. So by applying Lemma E.1.3, it is the case that each entry $|Z_i^{(k)*}| = |\hat{\mu}_D(z_i^*)|^k \leq (1 + r^D)^D \leq 1 + 2^{-D}$ for all $k \leq \log_2 D$.

Now, for an input $\xi \in \mathbb{C}$ we define the map

$$\rho^*(\xi) = \frac{-4N^2 r^D + \xi}{\|g\|_{\mathcal{A}}}, \tag{196}$$

and it's easy to confirm that we exactly represent:

$$g'(X) = \rho^* \left( \sum_{n,n'=1}^{2N} \psi_1^*(x_n, x_n') \right). \tag{197}$$

## F.2  Approximate Representation

Now, we can imitate the network above using the exp activation, and control the approximation error in the infinity norm. Let us assume we've chosen $f_1, f_2$ as in Lemma G.3. Furthermore, let us define $\xi \star \omega = \frac{1}{2} (f_1(\xi + \omega) - f_1(\xi) - f_1(\omega))$, so that $\star$ approximates scalar multiplication.

Then we mimic the exact network via:

$$z = (x_1 \star x_1', \ldots, x_D \star x_D') \tag{198}$$

$$Z^{(1)} = (f_2(z_1), \ldots, f_2(z_D)) \in \mathbb{C}^D \tag{199}$$

$$Z^{(2)} = \left( Z_1^{(1)} \star Z_2^{(1)}, \ldots, Z_{D-1}^{(1)} \star Z_D^{(1)} \right) \in \mathbb{C}^{D/2} \tag{200}$$

$$\ldots \tag{201}$$

$$Z^{(\log_2 D)} = Z_1^{(\log_2 D - 1)} \star Z_2^{(\log_2 D - 1)} \in \mathbb{C} \tag{202}$$

$$\psi_1(x, x') = Z^{(\log_2 D)}. \tag{203}$$

In other words, we replace all instances of multiplication $\cdot$ with $\star$, and all instances of $\hat{\mu}_D$ with $f_2$. Finally, we define the map $\rho$ as:

$$\rho(\xi) = \frac{4N^2}{\|g\|_{\mathcal{A}}} \cdot \left( \frac{\xi}{4N^2} \star 1 - r^D \right), \tag{204}$$

where we can clearly represent the constant $r^D$ via one additional neuron.

## F.3  Proof of Upper Bound

We complete the approximation of $g'$ by showing the exact and approximate networks are nearly equivalent in infinity norm, leveraging the assumption on our activation.

**Theorem F.1.** *Consider $\epsilon > 0$ such that $\epsilon \leq \min\left(\frac{1}{100}, \frac{1}{12D^2}\right)$. For $L = 1$, there exists $f \in Sym_L^2$, parameterized with an activation $\sigma$ that satisfies Assumption C.1, with width $O(D^3 + D^2 \log \frac{DN}{\epsilon})$, depth $O(\log D)$, and maximum weight magnitude $D \log D$ such that over inputs $X \in \mathbb{C}^{D \times 2N}$ with unit norm entries:*

$$\|f - g'\|_\infty \leq \epsilon. \tag{205}$$

*Proof.* Let $f$ be given by the $Sym_L^2$ network calculated in the previous section, i.e.

$$f(X) = \rho \left( \sum_{n,n'=1}^{2N} \psi_1(x_n, x_n') \right). \tag{206}$$

Clearly $L = 1$. From Assumption C.1 and what it guarantees about $f_1$ and $f_2$, it's clear that the maximum width of $f$ is $O(D^3 + D^2 \log \frac{D}{\epsilon})$, the depth is $O(\log D)$, and the maximum weight magnitude is $O(D \log D)$.

We can prove the quality of approximation by matching layer by layer. First we note a quick lemma:

**Lemma F.2.** *For $|\xi|, |\omega| \leq \frac{3}{2}$:*

$$|\xi \star \omega - \xi \cdot \omega| \leq \frac{3}{2}\epsilon \,. \tag{207}$$

*Proof.* Based on Assumption C.1, note that for $|\xi|, |\omega| \leq \frac{3}{2}$, we have that $|\xi + \omega| \leq 3$ and therefore:

$$|\xi \star \omega - \xi \cdot \omega| \leq \frac{1}{2}\left(|f_1(\xi + \omega) - (\xi + \omega)^2| + |f_1(\xi) - \xi^2| + |f_1(\omega) - \xi^2|\right) \tag{208}$$

$$\leq \frac{3}{2}\epsilon \,. \tag{209}$$

$\square$

It follows that, because all $|x_i| = 1$:

$$\|z^* - z\|_\infty = \max_{i \leq D} |x_i \star x_i' - x_i \cdot x_i'| \leq \frac{3}{2}\epsilon \,. \tag{210}$$

Now, because $|z_i^*| = 1$, it follows from our assumption on $\epsilon$ that $|z_i| \leq 1 + \frac{3}{2}\epsilon \leq 1 + \frac{1}{D}$. Hence, we can apply Lemma E.1.6 and say

$$\|Z^{(1)*} - Z^{(1)}\|_\infty = \max_{i \leq D} |\hat{\mu}_D(z_i^*) - f_2(z_i)| \tag{211}$$

$$\leq \max_{i \leq D} |\hat{\mu}_D(z_i^*) - \hat{\mu}_D(z_i)| + |\hat{\mu}_D(z_i) - f_2(z_i)| \tag{212}$$

$$\overset{(a)}{\leq} 6\left(\frac{3}{2}\epsilon\right) + \epsilon \tag{213}$$

$$\leq 10\epsilon \,. \tag{214}$$

where $(a)$ follows from Lemma E.1.6 and Assumption C.1 again.

Note, observe the following inequality, for any $i$:

$$|Z_{2i}^{(1)*} \cdot Z_{2i+1}^{(1)*} - Z_{2i}^{(1)} \cdot Z_{2i+1}^{(1)}| \leq |Z_{2i}^{(1)*} \cdot Z_{2i+1}^{(1)*} - Z_{2i}^{(1)*} \cdot Z_{2i+1}^{(1)}| + |Z_{2i}^{(1)*} \cdot Z_{2i+1}^{(1)} - Z_{2i}^{(1)} \cdot Z_{2i+1}^{(1)}| \tag{215}$$

$$= |Z_{2i}^{(1)*}| \cdot |Z_{2i+1}^{(1)*} - Z_{2i+1}^{(1)}| + |Z_{2i+1}^{(1)}| \cdot |Z_{2i}^{(1)*} - Z_{2i}^{(1)}| \tag{216}$$

$$= |\hat{\mu}_D(z_{2i}^*)| \cdot 10\epsilon + |f_2(z_{2i+1})| \cdot 10\epsilon \tag{217}$$

$$\overset{(a)}{\leq} 10\epsilon(|\hat{\mu}_D(z_{2i}^*)| + |\hat{\mu}_D(z_{2i+1})| + \epsilon) \tag{218}$$

$$\overset{(b)}{\leq} 10\epsilon\left(|\hat{\mu}_D(z_{2i}^*)| + |\hat{\mu}_D(z_{2i+1}^*)| + 6\left(\frac{3}{2}\epsilon\right) + \epsilon\right) \tag{219}$$

$$\overset{(c)}{\leq} 10\epsilon(1 + r^D + 1 + r^D + 4\epsilon + \epsilon) \tag{220}$$

$$\overset{(d)}{\leq} 10\epsilon(5/2) \tag{221}$$

$$\leq 25\epsilon \,, \tag{222}$$

where $(a)$ follows from Lemma G.3, $(b)$ follows from Lemma E.1.6, $(c)$ follows from Lemma E.1.3, and $(d)$ follows from the fact that $\epsilon \leq \frac{1}{100}$.

Hence, to draw error bounds one layer higher, we calculate:

$$\|Z^{(2)*} - Z^{(2)}\|_\infty = \max_{i \le D/2} |Z_{2i}^{(1)*} \cdot Z_{2i+1}^{(1)*} - Z_{2i}^{(1)} \star Z_{2i+1}^{(1)}| \tag{223}$$

$$\le \max_{i \le D/2} |Z_{2i}^{(1)*} \cdot Z_{2i+1}^{(1)*} - Z_{2i}^{(1)} \cdot Z_{2i+1}^{(1)}| + |Z_{2i}^{(1)} \cdot Z_{2i+1}^{(1)} - Z_{2i}^{(1)} \star Z_{2i+1}^{(1)}| \tag{224}$$

$$\overset{(a)}{\le} 25\epsilon + \frac{3}{2}\epsilon \tag{225}$$

$$\le 27\epsilon \,, \tag{226}$$

where in line $(a)$ we apply Lemma F.2 under the assumption that $|Z_i^{(1)}| \le \frac{3}{2}$ for all $i$.

Note that from Lemma E.1.3

$$|Z_i^{(1)}| \le |Z_i^{(1)} - Z_i^{(1)*}| + |Z_i^{(1)*}| \tag{227}$$

$$\le 10\epsilon + 1 + r^D < \frac{3}{2} \tag{228}$$

so this assumption is guaranteed.

We induct upwards through layers: assume that $\|Z^{(k)*} - Z^{(k)}\|_\infty \le 3^{k+1}\epsilon$ for $k \ge 2$. Then:

$$|Z_{2i}^{(k)*} \cdot Z_{2i+1}^{(k)*} - Z_{2i}^{(k)} \cdot Z_{2i+1}^{(k)}| \le |Z_{2i}^{(k)*} \cdot Z_{2i+1}^{(k)*} - Z_{2i}^{(k)*} \cdot Z_{2i+1}^{(k)}| + |Z_{2i}^{(k)*} \cdot Z_{2i+1}^{(k)} - Z_{2i}^{(k)} \cdot Z_{2i+1}^{(k)}| \tag{229}$$

$$= |Z_{2i}^{(k)*}| \cdot |Z_{2i+1}^{(k)*} - Z_{2i+1}^{(k)}| + |Z_{2i+1}^{(k)}| \cdot |Z_{2i}^{(k)*} - Z_{2i}^{(k)}| \tag{230}$$

$$\overset{(a)}{\le} 3^{k+1}\epsilon(|Z_{2i}^{(k)*}| + |Z_{2i+1}^{(k)}|) \tag{231}$$

$$\overset{(b)}{\le} 3^{k+1}\epsilon(|Z_{2i}^{(k)*}| + |Z_{2i+1}^{(k)*}| + 3^{k+1}\epsilon) \tag{232}$$

$$\overset{(c)}{\le} 3^{k+1}\epsilon((1 + r^D)^D + (1 + r^D)^D + 3^{k+1}\epsilon) \tag{233}$$

$$\overset{(d)}{\le} 3^{k+1}\epsilon\left(1 + 2^{-D} + 1 + 2^{-D} + \frac{1}{4}\right) \tag{234}$$

$$\le 3^{k+1}\epsilon\left(\frac{11}{4}\right) \,, \tag{235}$$

where $(a)$ and $(b)$ are both applications of the inductive hypothesis, $(c)$ follows from Lemma E.1.3, $(d)$ is the binomial inequality and the fact that for any $k \le \log_2 D$:

$$3^{k+1}\epsilon \le 3\left(4^{\log_2 D}\right)\epsilon \tag{236}$$

$$= \frac{\epsilon}{3D^2} \tag{237}$$

$$\le \frac{1}{4} \,. \tag{238}$$

And as before:

$$\|Z^{(k+1)*} - Z^{(k+1)}\|_\infty = \max_i |Z_{2i}^{(k)*} \cdot Z_{2i+1}^{(k)*} - Z_{2i}^{(k)} \star Z_{2i+1}^{(k)}| \tag{239}$$

$$\le \max_i |Z_{2i}^{(k)*} \cdot Z_{2i+1}^{(k)*} - Z_{2i}^{(k)} \cdot Z_{2i+1}^{(k)}| + |Z_{2i}^{(k)} \cdot Z_{2i+1}^{(k)} - Z_{2i}^{(k)} \star Z_{2i+1}^{(k)}| \tag{240}$$

$$\overset{(a)}{\le} 3^{k+1}\epsilon\left(\frac{11}{4}\right) + \frac{3}{2}\epsilon \tag{241}$$

$$\le 3^{k+2}\epsilon \,, \tag{242}$$

where in line $(a)$ we apply Lemma F.2 under the assumption that $|Z_i^{(k)}| \le \frac{3}{2}$ for all $i$.

Note that as before

$$|Z_i^{(k)}| \leq |Z_i^{(k)} - Z_i^{(k)*}| + |Z_i^{(k)*}| \tag{243}$$

$$\leq 3^{k+1}\epsilon + (1 + r^D)^D \tag{244}$$

$$\leq 3^{k+1}\epsilon + 1 + 2^{-D} \leq \frac{3}{2} , \tag{245}$$

so the assumption is granted.

Thus, completing the induction and remembering the definition of $\psi_1$, we conclude:

$$\|\psi_1^*(x_n, x_{n'}) - \psi_1(x_n, x_{n'})\|_\infty \leq 3^{\log_2 D + 1}\epsilon < 3D^2\epsilon . \tag{246}$$

Hence, we can finally bound the final networks:

$$\|g' - f\|_\infty = \left\| \rho^* \left( \sum_{n,n'=1}^{2N} \psi_1^*(x_n, x_{n'}) \right) - \rho \left( \sum_{n,n'=1}^{2N} \psi_1(x_n, x_{n'}) \right) \right\|_\infty \tag{247}$$

$$= \frac{1}{\|g\|_\mathcal{A}} \left\| \sum_{n,n'=1}^{2N} \psi_1^*(x_n, x_{n'}) - 4N^2 \left( \left[ \frac{1}{4N^2} \sum_{n,n'=1}^{2N} \psi_1(x_n, x_{n'}) \right] \star 1 \right) \right\|_\infty \tag{248}$$

$$\overset{(a)}{\leq} 4N^2 \left\| \frac{1}{4N^2} \sum_{n,n'=1}^{2N} \psi_1^*(x_n, x_{n'}) - \left( \left[ \frac{1}{4N^2} \sum_{n,n'=1}^{2N} \psi_1(x_n, x_{n'}) \right] \star 1 \right) \right\|_\infty \tag{249}$$

$$\overset{(b)}{\leq} 4N^2 \left\| \frac{1}{4N^2} \sum_{n,n'=1}^{2N} \psi_1^*(x_n, x_{n'}) - \frac{1}{4N^2} \sum_{n,n'=1}^{2N} \psi_1^*(x_n, x_{n'}) \right\|_\infty + 4N^2 \cdot \frac{3}{2}\epsilon \tag{250}$$

$$\leq 4N^2 \|\psi_1^*(x, x') - \psi(x, x')\|_\infty + 4N^2 \cdot \frac{3}{2}\epsilon \tag{251}$$

$$\leq 12N^2 D^2 \epsilon + 6N^2 \epsilon \tag{252}$$

$$\leq 18N^2 D^2 \epsilon , \tag{253}$$

where in $(a)$ we apply the lower bound $\|g\|_A \geq 1$ from E.3.2 and in $(b)$ we once again apply Lemma F.2, valid from the fact that for all $X$ with unit norm entries:

$$\left| \frac{1}{4N^2} \sum_{n,n'=1}^{2N} \psi_1(x_n, x_{n'}) \right| \leq 3D^2\epsilon \leq \frac{3}{2} . \tag{254}$$

So it remains to map $\epsilon \to \frac{\epsilon}{18N^2 D^2}$ in order to yield that $\|f - g'\| \leq \epsilon$. Note that this remapping only changes the maximum width to be $O(D^3 + D^2 \log \frac{ND}{\epsilon})$. $\qquad\square$

# G  Activation Assumption for $\exp$

We prove that the activation $\exp$ satisfies Assumption C.1.

We need the following standard fact, whose proof we include for completeness:

**Lemma G.1.** *Fix $J$ and let $\gamma$ be a primitive $J$th root of unity. Then*

$$\frac{1}{J} \sum_{j=0}^{J-1} \gamma^{ij} = \begin{cases} 1 & i \equiv 0 \mod J \\ 0 & i \not\equiv 0 \mod J \end{cases} \tag{255}$$

*Proof.* If $i \equiv 0 \mod J$, then $\gamma^{ij} = 1$ for all integer $j$ and clearly

$$\frac{1}{J} \sum_{j=0}^{J-1} \gamma^{ij} = 1 . \tag{256}$$

Suppose $i \not\equiv 0 \mod J$. Note that any $J$th root of unity $x$ must satisfy $x^J = 1$, or equivalently

$$(1 - x) \left( \sum_{j=0}^{J-1} x^j \right) = 0 . \tag{257}$$

Because $i \not\equiv 0 \mod J$ and $\gamma$ is a primitive root, it follows $\gamma^i \neq 1$ is another root. Therefore setting $x = \gamma^i$ and factoring out the non-zero term $(1 - \gamma^i)$ gives

$$\sum_{j=0}^{J-1} \gamma^{ij} = 0 . \tag{258}$$

$\square$

Using this fact, we can approximate simple analytic functions via shallow networks in the $\exp$ activation.

**Lemma G.2.** *For any $J \in \mathbb{N}$ with $J > D$, there exists a shallow neural networks $f_1, f_2$ using the $\exp$ activation, with $O(JD)$ neurons and $O(D \log D)$ weights, such that*

$$\sup_{|\xi| \leq 3} \left| f_1(\xi) - \xi^2 \right| \leq \frac{4}{J!} \left( \frac{3}{4} \right)^J \tag{259}$$

$$\sup_{|\xi| \leq 3} |f_2(\xi) - \hat{\mu}_D(\xi)| \leq 17D \left( \frac{3}{4} \right)^J . \tag{260}$$

*Proof.* Let $\gamma$ be a primitive $J$th root of unity, $r = 1/4$, and let $k \in \mathbb{N}$ such that $0 \leq k \leq J - 1$. By applying Lemma G.1 we can define a network $f^{(k)}$ and expand as:

$$f^{(k)}(\xi) := \sum_{j=0}^{J-1} \frac{\gamma^{-kj}}{J} \exp(\gamma^j r \xi) \tag{261}$$

$$= \sum_{j=0}^{J-1} \frac{\gamma^{-kj}}{J} \sum_{i=0}^{\infty} \frac{(\gamma^j r \xi)^i}{i!} \tag{262}$$

$$= \sum_{i=0}^{\infty} \frac{(r\xi)^i}{i!} \left[ \frac{1}{J} \sum_{j=0}^{J-1} \gamma^{(i-k)j} \right] \tag{263}$$

$$= \sum_{i=0}^{\infty} \frac{(r\xi)^i}{i!} \mathbb{1}_{i \equiv k \mod J} \tag{264}$$

$$= \sum_{i=0}^{\infty} \frac{(r\xi)^{iJ+k}}{(iJ + k)!} \tag{265}$$

$$= \frac{(r\xi)^k}{k!} + \sum_{i=1}^{\infty} \frac{(r\xi)^{iJ+k}}{(iJ + k)!} . \tag{266}$$

It follows that we can bound:

$$\sup_{|\xi|\leq 3}\left|f^{(k)}(\xi)-\frac{(r\xi)^k}{k!}\right|\leq\sum_{i=1}^{\infty}\left|\frac{(r\xi)^{iJ+k}}{(iJ+k)!}\right| \tag{267}$$

$$\leq\frac{1}{J!}\sum_{i=1}^{\infty}\left(\frac{3}{4}\right)^{iJ+k} \tag{268}$$

$$\leq\frac{1}{J!}\left(\frac{3}{4}\right)^{J}\sum_{i=0}^{\infty}\left(\frac{3}{4}\right)^{iJ} \tag{269}$$

$$\leq\frac{1}{J!}\left(\frac{3}{4}\right)^{J}\frac{1}{1-(3/4)^J} \tag{270}$$

$$\leq\frac{4}{J!}\left(\frac{3}{4}\right)^{J}, \tag{271}$$

so we can define

$$f_1(\xi):=\frac{2}{r^2}f^{(2)}(\xi) \tag{272}$$

with only $J$ neurons each of width magnitude at most $O(1)$, and instantly gain the bound

$$\sup_{|\xi|\leq 3}\left|f_1(\xi)-\xi^2\right|=\sup_{|\xi|\leq 3}\frac{2}{r^2}\left|f^{(k)}(\xi)-\frac{(r\xi)^2}{2!}\right| \tag{273}$$

$$\leq\frac{2}{r^2}\cdot\frac{4}{J!}\left(\frac{3}{4}\right)^{J}. \tag{274}$$

Second, we define

$$f_2(\xi):=r\left(\sum_{k=0}^{D-1}k!f^{(k)}(\xi)\right)-\sum_{k=1}^{D}\frac{k!}{r}f^{(k)}(\xi). \tag{275}$$

First, let us note that, in spite of seeming to have factorial weights, we can write this network with small weights via properties of the exponential:

$$f_2(\xi)=r\left(\sum_{k=0}^{D-1}\exp(\log k!)f^{(k)}(\xi)\right)-\sum_{k=1}^{D}\frac{\exp(\log k!)}{r}f^{(k)}(\xi) \tag{276}$$

$$=r\sum_{k=0}^{D-1}\sum_{j=0}^{J-1}\frac{\gamma^{-kj}}{J}\exp(\log k!+\gamma^j r\xi)-\sum_{k=1}^{D}\frac{1}{r}\sum_{j=0}^{J-1}\frac{\gamma^{-kj}}{J}\exp(\log k!+\gamma^j r\xi). \tag{277}$$

The network contains $2DJ$ neurons, with the norm of each weight bounded by $O(D\log D)$.

Then using the decomposition

$$\hat{\mu}_D(\xi)=r\sum_{k=0}^{D-1}(r\xi)^k-\frac{1}{r}\sum_{k=1}^{D}(r\xi)^k$$

we derive:

$$\sup_{|\xi|\leq 3}|f_2(\xi)-\hat{\mu}_D(\xi)|\leq\sup_{|\xi|\leq 3}\left|r\left(\sum_{k=0}^{D-1}k!f^{(k)}(\xi)\right)-r\sum_{k=0}^{D-1}(r\xi)^k\right|+\left|\left(\sum_{k=1}^{D}\frac{k!}{r}f^{(k)}(\xi)\right)-\frac{1}{r}\sum_{k=1}^{D}(r\xi)^k\right| \tag{278}$$

$$\leq\left(\sum_{k=0}^{D-1}rk!\right)\frac{4}{J!}\left(\frac{3}{4}\right)^{J}+\left(\sum_{k=1}^{D}\frac{k!}{r}\right)\frac{4}{J!}\left(\frac{3}{4}\right)^{J} \tag{279}$$

$$\leq 17D\left(\frac{3}{4}\right)^{J}. \tag{280}$$

$\square$

Now, let us restate this result, choosing the error rate $\epsilon$ explicitly:

**Lemma G.3.** *For any $\epsilon > 0$, there exists a shallow neural networks $f_1$, $f_2$ using the* exp *activation, with $O\left(D^2 + D\log\frac{D}{\epsilon}\right)$ neurons and $O(D\log D)$ weights, such that*

$$\sup_{|\xi|\leq 3} \left|f_1(\xi) - \xi^2\right| \leq \epsilon\,, \tag{281}$$

$$\sup_{|\xi|\leq 3} |f_2(\xi) - \hat{\mu}_D(\xi)| \leq \epsilon\,. \tag{282}$$

We remark again that, in the event $D > \sqrt{N/2}$, we replace $D$ with $\hat{D}$ in order to approximate the Blaschke product $\hat{\mu}_{\hat{D}}$ as this is the function we use to build the hard function $g$ in that case. So we recover the statement of Assumption C.1.