# OpenReview forum: "Exponential Separations in Symmetric Neural Networks"
_NeurIPS.cc/2022/Conference — NeurIPS 2022 Accept_

### Official Review · Reviewer_WQz6 · 2022-07-09

**Rating:** 7
**Confidence:** 3
**Soundness:** 4 excellent
**Presentation:** 4 excellent
**Contribution:** 3 good

**Summary:**

This paper provides a novel result on the relative expressiveness of two popular architectures for symmetric functions, Deep Sets (pooling over individual inputs) and Relational Networks (pooling over pairs of inputs).

**Questions:**

The main result in this paper shows that there exists a "hard" function $g$ for which Deep Sets requires an exponentially greater symmetric width than Relational Networks. Is your lower bound on the symmetric width for Deep Sets the worst possible, i.e. does the symmetric width required by your function $g$ suffice for Deep Sets to model any symmetric function? Is there a known bound on the required symmetric width for Relational Networks to model any symmetric function?


**Limitations:**

I see no potential for negative societal impact. The authors briefly note a major limitation of their result. I think that this brief note is adequate, since it correctly identifies the most important possible strengthening of the result and explicitly leaves it open for future work.


**Strengths And Weaknesses:**

This is a very well-written paper. The narrative is easy to follow and the key ideas are explained clearly. The key result is novel and interesting, and motivation is provided in the text. This paper will certainly be of significant interest to researchers working with symmetric neural networks.

Most of the paper is devoted to setting up mathematical context or describing the mathematics of proving the main result. The mathematical content here is explained and structured very clearly, but perhaps a small amount of additional space could be given over to motivation and practical context for the result, which, although present, is currently a little lacking.

---

> ### Author Response · Authors · 2022-07-29
> **Rebuttal**
>
> Thanks for your review!  We address your concerns and questions below:
>
> We appreciate the feedback that motivation was insufficient and we agree, see the response to reviewer YvXW for planned changes to explain why this separation is natural and practically motivated.
>
> The tightness of our lower bound is a good question; in the D=1 case, the symmetric width is tight up to a factor of 2.  In the D>1 case, there is a non-trivial gap between the symmetric width required to model our hard function and the width required to model any function. We will add a discussion on this interesting point in the updated version.
>
> To the best of our knowledge, there is no known symmetric width bound for Relational Networks, beyond trivially inheriting the bounds on Deepsets.

---

### Official Review · Reviewer_c2Zx · 2022-07-12

**Rating:** 7
**Confidence:** 3
**Soundness:** 3 good
**Presentation:** 3 good
**Contribution:** 3 good

**Summary:**

The paper studies expressivity capabilities of neural networks that belong to the class of symmetric neural networks. Two most prominent examples that are used as motivation for the current work is the so-called DeepSets and Relational Networks. Even though these are universal approximators for representing symmetric functions, what are the depth/width tradeoffs and approximation guarantees? This question reflects the analogous questions for standard feedforward neural networks.

The main contribution of the paper is to explicitly construct symmetric functions which provably require exponentially-many neurons in the
DeepSets model, yet are efficiently approximated with self-attention. The crucial parameter controling the expressivity of the network here, is the so-called "symmetric width" and this leads to a conceptually different set of results in comparison to standard depth vs width tradeoffs.

The difference in the two architectures is presented by eq. (2) vs eq. (4), where the latter allows for pairwise interactions among set elements.

Under plausible assumptions, the main result of the paper is formally Theorem 2.4 is to provide a family of analytic symmetric functions g which leads to two important properties in order to prove separation results. The first part says that "singleton symmetric networks" (i.e., those that don't allow for pairwise interactions) are insufficient to approximate g unless they are exponentially large in terms of their symmetric width, whereas the second part says that a simple "pairwise symmetric network" will incur negligible loss when trying to approximate g. The notions of error used are: for the lower bound in the first part, the authors used the notion of L2 error under a suitable data distribution, and for the second part in the upper bound they use the infinity norm error.

**Questions:**

-complex values vs real values? One question that comes up has to do with the fact that the input is complex-valued. How does this affect the statement of the result for real-valued neural nets? Ie does the construction break down if we only cared about real valued networks?

-omissions in literature/failed to do comparisons:
1. In Chulhee Yun et al. ("Are Transformers universal approximators of sequence-to-sequence functions?") the authors study sequence-to-sequence models and ask about the approximation/representation capabilities. Why did the authors fail to compare against this paper, given that Set Transformers  are special instantiations of Transformers?
2. There are also works in expressivity that manage to show separations based on topological properties of the function-to-be-represented. These hold for feedforward neural nets and use zonotope theory (see UNDERSTANDING DEEP NEURAL NETWORKS WITH
RECTIFIED LINEAR UNITS by Arora et al.) or fixed-point arguments in dynamical systems (see Better Depth-Width Trade-offs for Neural Networks through the lens of Dynamical Systems by Chatziafratis et al.). Is there hope to use approach taken there to show separations for symmetric nets? What is the crucial property of symmetric nets being exploited in the present paper? Is the main observation of the paper the fact that the pairwise interaction is a "complexity" measure of sorts, that cannot be simulated by the "singleton symmetric net"?
3. Can the authors elaborate more on why the L2 error lower bound was used in eq. (6) rather than the stronger lower bound of L1? From Telgarsky's work cited (Benefits of depth in neural networks) it seems that L1 is the stronger guarantee. What is the main technical bottleneck for obtaining L2? Thank you!




**Strengths And Weaknesses:**

-solid contribution for theory of expressivity in neural nets that offers new perspective and techniques to a different set of architectures.
-conceptually interesting the fact that the result bears differences with standard results of feedforward neural nets separations

Weaknesses:
-limited literature comparisons e.g., Chulhee Yun et al. ("Are Transformers universal approximators of sequence-to-sequence functions?")?
-L2 error bound instead of L1 error bound?
-please see questions below.

Overall, the reviewer thinks that the paper has to offer sth new to the much needed theory of neural nets for symmetic function representation and that the paper does so in a technically solid manner.

---

> ### Author Response · Authors · 2022-07-29
> **Rebuttal**
>
> Thanks for your review, please see the response to your questions below:
>
> It is true that our construction relies on complex inputs owing to the use of the Hall inner product.  It is a relevant open question whether the result can be extended to the real-valued case.
>
> The paper "Are Transformers universal approximators of sequence-to-sequence functions?") is related to our discussion of Set Transformers in the adjacent realm of permutation equivariant universal approximation.  But it doesn't propose a substantially different architecture than Set Transformers, which is primarily a transformer without positional encodings.  The question of extending our analysis to cover the transformer architecture of both these papers is still open.
>
> The first paper you mention, "Understanding Deep Neural Networks with Rectified Linear Units", relies on topological arguments with some similarity to those in [1], which builds a hard function for DeepSets in the D = 1 case, though their result does not require exponential symmetric width and only holds in the infinity norm.  Whether those ideas, or the dynamical system ideas, can be adapted for symmetric networks is not at all clear to us.  The crucial property we exploit of symmetric networks is that, by their design, they can be written in terms of powersum polynomials which enjoy certain orthogonality properties.  And yes, the main observation of the paper is that certain functions are too complicated, as measured by the rank of an induced matrix, to be written efficiently without pairwise interaction.
>
> For our proof method, we use L2 norm because we calculate the norm using an orthogonal decomposition of symmetric polynomials.  Of course this method would not hold for an L1 norm, and it's a relevant open question what kinds of L1 norm guarantees can be obtained.
>
> [1] Wagstaff, Edward, et al. "Universal approximation of functions on sets." Journal of Machine Learning Research 23.151 (2022): 1-56.

---

### Official Review · Reviewer_Ddfp · 2022-07-12

**Rating:** 8
**Confidence:** 3
**Soundness:** 4 excellent
**Presentation:** 4 excellent
**Contribution:** 4 excellent

**Summary:**

This paper considers the representational power of neural network functions. It considers two types of network architectures: the DeepSets architecture, which treats the inputs in a permutation invariant manner; and the Relational Network architecture, which allows for pairwise interaction among the inputs. The question considered in this paper is to compare the representation power of these two types of architectures.

The main result is a "width separation" between the DeepSets and the Relational Network architectures. It shows that there exists a function such that
- for any width less than $exp(min(input\ dimension, \sqrt{input\ set\ size})$, the best function from the DeepSets family incurs a constant error.
- on the other hand, with $poly(input\ dimension, input\ set\ size)$ width, the Relational Network could represent the same function up to arbitrary small error.

The lower bound for the one-dimensional case is as follows:
- the "DeepSets" networks with width $L$ restricts the space of functions to within some space of rank relating to $L$; so for a "high" rank function, its orthogonal projection to this rank $L$ space could still be quite large

The high-dimensional case requires using a high-dimensional powersum polynomials. However, the construction of the lower bound instance is more delicate than the above.

**Questions:**

- What are some examples of activation functions under the analytic assumption?
- For the high-dimensional case, the construction of the lower bound instance (in section 5.3) seems to require much more work; A better explanation of this part would be appreciable.
- Almost all of the references are shown in an et al. style on page 10 and 11, except a few. Is this done on purpose, e.g., should the names of all authors be properly cited there?

**Limitations:**

Both the limitations and the potential negative societal impact are discussed in the paper.

**Strengths And Weaknesses:**

**Strength**

- This paper deals with an important yet technically challenging question. The proof requires sophisticated machinery from symmetric polynomial theory. However, the authors do a great job of explaining their results and building up their proofs, which is instructive for a reader.
- The writing of the paper is a pleasure to read, with particular attention paid to the proof details.
- The width separation between DeepSets and Relational Networks would be a significant contribution to the literature; for example, one could imagine that the machinery developed here might help separate architectures in other settings such as graph neural networks, where this sort of permutation invariance appears quite often.

**Weakness**

- The main result relies on a certain analytic assumption on the activation functions of the neural network (as the authors have discussed in the limitations section).

---

> ### Author Response · Authors · 2022-07-29
> **Rebuttal**
>
> Thank you for your review, we address your questions below:
>
> Examples of analytic activations include exp, cosh, and tanh when restricted away from its poles.  In practice this can be achieved by clipping weights to make sure inputs never approach the poles.
>
> We agree the lower bound deserve more elaboration, and will update a later draft to smooth the characterization of the lower bound function.
>
> Thanks for catching this error in our references, a flag was set wrong in our bibtex.  It now correctly prints all authors up to four, and only uses et al. for five or above.

---

### Official Review · Reviewer_YvXW · 2022-07-13

**Rating:** 7
**Confidence:** 4
**Soundness:** 4 excellent
**Presentation:** 3 good
**Contribution:** 3 good

**Summary:**

The paper studies the expressive power of symmetric neural networks used to compute functions on sets, specifically Relational Networks and DeepSets architectures. An exponential separation result is given where a certain function can be represented efficiently using a Relational Network with symmetric width L=1, yet cannot be approximated by a DeepSet architecture to accuracy better than a constant unless its symmetric width is exponential in either of $N$, the size of the set; or $\sqrt{D}$, where $D$ is the dimension of each element.

**Questions:**

- Can you motivate the study of this problem further? I'm curious to know if similarly to Eldan and Shamir, there are known practical problems where Relational Networks outperform DeepSets. Such examples will make the paper better motivated in my opinion.

- Are complex networks used in practice in any context? This will make the assumption on using complex numbers much more justified.

- In the paper, you clearly state the analyticity assumption as a weakness in your paper. You do not discuss, however, what commonly used activations satisfy your assumption. Is Assumption 2.3 satisfied by any sigmoidal activation, e.g. logistic, tanh, or arctan? Providing an example beyond the exponential activation will make this weakness much more mild.

**Limitations:**

Yes

**Strengths And Weaknesses:**

Strengths:

- The paper is well-written and easy to follow.

- The proof technique used bears a certain (subjective) elegance. The mathematical tools used in the proof which leverage symmetric polynomials to establish the main result seem like the natural machinery for studying the expressive power of symmetric architectures, and the proof idea seems like a clever use of these tools.



Weaknesses:

- It feels like the paper does not motivate the study of the problem addressed in it sufficiently. To give a concrete comparison, the separation in [1] is well-motivated since it was widely observed that deeper architectures are much more successful in many learning tasks. It is not clear if this is also the case when using Relational Networks instead of DeepSets.

- The proof technique seems to make an essential use of complex numbers and analytic activations. While the authors are overall transparent about these weaknesses, it doesn't feel like they are properly discussed.

- It was shown in [2] that certain functions used to separate depth cannot be learned using gradient methods. Since the construction used to approximate the target function using a Relational Network seems intricate and also uses a relatively deep architecture, I'm not sure if it can be learned efficiently. While I find the result in the paper interesting nevertheless, I think it would be appropriate to add a comment about this.


Citations used:

[1] - Eldan and Shamir: The Power of Depth for Feedforward Neural Networks

[2] - Malach et al.: The Connection Between Approximation, Depth Separation and Learnability in Neural Networks

---

> ### Author Response · Authors · 2022-07-29
> **Rebuttal**
>
> Thank you for your thoughtful review!  We address your concerns and questions below:
>
> In terms of motivation, there are indeed several works that notice a performance difference in practice between Deepsets and models that enable higher order interaction.  [1] shows a substantial improvement when using a similar model to Relational Networks based on pairwise attention. [2] parameterizes functions in quantum chemistry using symmetric architectures and notes the performance drops substantially without pairwise features (see Table III).  [3] demonstrates the efficacy of a simplified parameterization of DeepSets vs a simplified Relational Network, corresponding to k-ary Janossy pooling nets for k = 1, 2.
>
> These works don't directly compare DeepSets vs Relational Networks. But in our view, because Relational Networks are the simplest model with interaction between set elements, they're poised as the natural choice to show a separation from Deepsets, especially because both models are used in practice quite often.  Relational Networks also serve as a strictly smaller class than Set Transformers when using arbitrarily expressive attention functions (the question of Set Transformers using dot-product attention remains open).  We will include this more rounded discussion of the motivation in an updated draft.
>
> We agree with the note that the complex analytic assumption and question of learnabiliity deserve more mention in the body, and will update this in the next draft.
>
> Complex networks do see practical use, see Table 4 in [4] for a survey of their application, primarily in signal processing and image processing.
>
> Based on analogy to the non-analytic case, we suspect any analytic, Lipschitz activation would meet our approximation assumptions, but are currently unable to prove this in general.  Nevertheless, we feel the main thrust of the paper is the existence of the separation under some activation, and leave the issue of proving this technical assumption is met for other activations to future work.
>
> [1] Lee, Juho, et al. "Set transformer: A framework for attention-based permutation-invariant neural networks." International conference on machine learning. PMLR, 2019.
>
> [2] Pfau, David, et al. "Ab initio solution of the many-electron Schrödinger equation with deep neural networks." Physical Review Research 2.3 (2020): 033429.
>
> [3] Murphy, Ryan L., et al. "Janossy pooling: Learning deep permutation-invariant functions for variable-size inputs." arXiv preprint arXiv:1811.01900 (2018).
>
> [4] Bassey, Joshua, Lijun Qian, and Xianfang Li. "A survey of complex-valued neural networks." arXiv preprint arXiv:2101.12249 (2021).

---

### Meta-Review · Area_Chair_FfBC · 2022-08-27

**Recommendation:** Accept
**Confidence:** Certain

**Metareview:**

Four domain experts recommended acceptance for this paper and I agree with their assessment. The writeup presents its setup, result, and argument all very clearly. The "width separation" result is solid and characterizes an exponential gap in expressive capacity between two material neural net architectures. Reviewers agree that the analysis carries independent interest/novelty as well, and are optimistic that it might be useful in other separation arguments (e.g. Reviewer Ddfp's comment about graph neural nets).

Especially in initial reviews, there were some concerns raised around motivation and grounding in practice. For instance, Reviewer YvXW naturally questioned the importance of separating the two types of architectures studied here, specifically asking whether this was grounded in a known empirical discrepancy between them. I think this was addressed well in the authors' subsequent response -- with references -- and it seems the reviewer agrees. Still, I found this thread helpful and I suspect that readers will naturally ask a similar question. I would recommend that the authors consider incorporating some of this reply into the paper itself. (Whether/how to do this is up to them, and doesn't bear on my acceptance recommendation.) In the same discussion, there was a good point raised to remark on learnability (e.g. by gradient descent). The authors mentioned they would comment on this in the next draft and I'd encourage that further as well.

Thanks to reviewers and authors both for their work. Overall this a nice research contribution and a well-written paper.

**Award:**

No

---

### Decision · Program_Chairs · 2022-09-14

Accept